# ACTIVATION WITH INTRINSIC-EXTRINSIC CONSENSUS

## ABSTRACT

Artificial Neural Networks (ANNs) are powerful tools for complex pattern recognition and decision-making. While existing activation mechanisms often promote sparsity through thresholding, they lack an explicit assessment of channel relevance, making networks susceptible to interference from noisy channels. Such irrelevant activations can propagate through the network and adversely affect the final decision. Inspired by observations that channel relevance can be assessed from both intrinsic activity levels and extrinsic decision weights—and that a strong consensus exists between these two aspects—this paper proposes AIEC (Activation with Intrinsic-Extrinsic Consensus), a novel activation mechanism designed to identify and suppress irrelevant channels during training. AIEC consists of three components: an intrinsic Activation-Counting Unit that tracks channel activation statistics, an extrinsic Decision-Making Unit that learns channel decision weights, and a Consensus Gatekeeping Unit that suppresses irrelevant channels based on the agreement between the intrinsic and extrinsic assessments. Extensive experiments demonstrate that AIEC effectively suppresses irrelevant channels and facilitates sparser neural representations. Furthermore, AIEC is compatible with a wide range of mainstream ANN architectures and achieves superior performance compared to existing activation mechanisms across multiple tasks and domains.

## 1 INTRODUCTION

Artificial Neural Networks (ANNs) (LeCun et al., 2015) have demonstrated remarkable capabilities in solving complex pattern recognition and decision-making problems. From original Multi-Layer Perceptron (MLP) (McClelland et al., 1987) to classical Convolutional Neural Networks (CNNs) (LeCun et al., 1998) and trendsetting Vision Transformers (ViTs) (Han et al., 2022), their evolving architectures have driven breakthroughs in computer vision (Voulodimos et al., 2018) and beyond (Zhou et al., 2022; Ren et al., 2023). A key factor behind their success lies in their feature learning. In particular, channel-wise features are crucial in semantic abstraction, forming the critical foundation for the networks to understand data and make decisions.

As a core component of ANNs' feature learning, activation mechanisms like ReLU (Glorot et al., 2011) employ thresholding to sparsify representations, achieving preliminary feature selection and offering advantages such as information disentanglement, linear separability, and potential generalization ability (Glorot et al., 2011). However, such mechanisms rely solely on instantaneous activation intensity and lack explicit assessment of channel relevance: on one hand, they struggle to distinguish transient noise from genuine features; on the other hand, they remain oblivious to channels' actual contributions to the network's decision. This limitation makes it difficult to effectively prevent noise interference from irrelevant channels. The erroneous activation of these channels may be amplified by subsequent layers, ultimately affecting the network's final decision. This highlights the urgent need for channel relevance assessment during the training process.

As observed in Figure 1, channel relevance can be measured from both intrinsic and extrinsic aspects. Intrinsically, under stimuli of homogeneous samples, some channels frequently activate and contribute to the network's decision, whereas the majority of channels, though rarely active, should ideally remain entirely silent. Extrinsically, the decision weights learned by a linear classifier acting on the post-activation feature vector can also reflect each channel's relevance to the final decision. Moreover, a strong consistency is observed between the intrinsic and extrinsic aspects, with both pointing to the same set of critical channels. This consensus can guide the network in assessing channel relevance throughout training and further suppressing interference from less relevant channels.

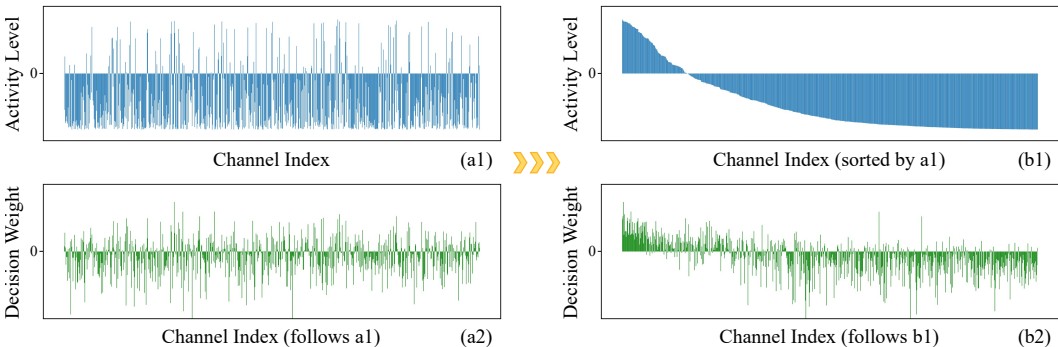

Figure 1: The intrinsic activity level (a1) and extrinsic decision weight (a2) are recorded for each channel of the feature vector after ReLU in the final block of ViT-Tiny, specifically for the "truck" category in the CIFAR-10 dataset. The activity level of a channel is computed based on its historical activation statistics over samples from the "truck" category, defined as "(number of activations - number of inhibitions) / total sample count". The decision weight is learned through a linear classifier applied to the post-activation feature vector. (b1) is (a1) sorted in descending order of activity level, and (b2) is (a2) re-indexed following the index order of (b1).

To address the lack of channel relevance assessment in existing activation mechanisms, and based on the above observations, we propose AIEC (Activation with Intrinsic-Extrinsic Consensus), a novel activation mechanism designed to identify and suppress irrelevant feature channels. AIEC integrates online assessment established through intrinsic-extrinsic consensus to identify irrelevant channels. At the intrinsic aspect, AIEC tracks threshold-activation statistics for each channel across different categories in real time, establishing a channel relevance assessment based on channels' activity levels. At the extrinsic aspect, AIEC learns decision weights for each channel through supervised feedback, constructing a channel relevance assessment based on channels' influence on the network's final decision. The final criterion for irrelevant channel identification is the consensus between the intrinsic and extrinsic assessments. Guided by this consensus, AIEC performs channel-wise gatekeeping for noise cleaning, effectively suppressing activation responses from irrelevant channels while preserving those from relevant channels. Moreover, the proposed AIEC is fast during both training and inference phases. Extensive experiments demonstrate that the proposed AIEC achieves outstanding performance compared to existing activation mechanisms across various mainstream ANN architectures, datasets, and multiple tasks and domains.

The contributions of this paper are summarized as follows:

- The observations that channel relevance can be assessed from two dimensions: intrinsic activity levels and extrinsic decision weights, and that a high consensus exists between them. This consensus provides a reliable basis for assessing channel relevance, an aspect overlooked by existing activation mechanisms.

- The proposal of AIEC (Activation with Intrinsic-Extrinsic Consensus), an innovative activation mechanism that identifies irrelevant channels based on the consensus of intrinsic and extrinsic aspects, and performs channel-wise gatekeeping for noise cleaning.

- Extensive experiments demonstrate that the proposed AIEC delivers outstanding performance compared to existing activation mechanisms across various mainstream ANN architectures, datasets, and multiple tasks and domains.

## 2 OBSERVATIONS

This section presents observations on ANN's channel activation from both intrinsic and extrinsic aspects, providing insights for addressing deficiencies in existing activation mechanisms.

***Observation 1:*** *Variance in channel activity levels intrinsically reflects channel relevance.*

Under threshold-based activation (e.g., ReLU), relevant signals get activated while irrelevant ones are inhibited. As illustrated in Figure 1(a1), when numerous homogeneous samples are input, different channels show different activity levels, which is deemed to be positively correlated with the channels'

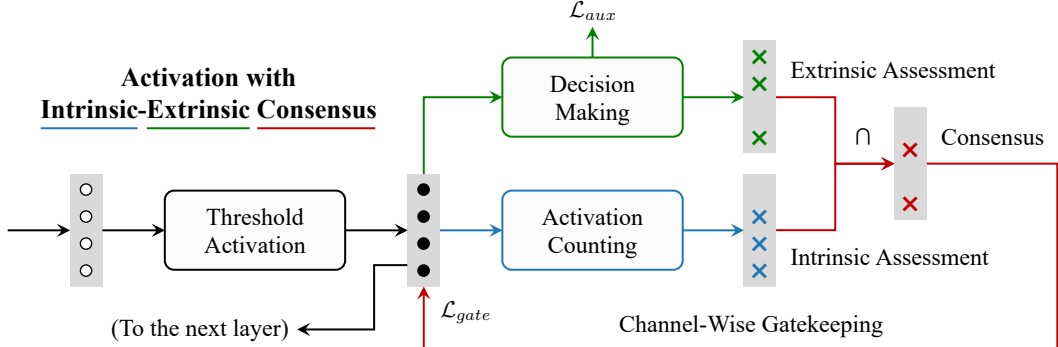

Figure 2: The proposed AIEC (Activation with Intrinsic-Extrinsic Consensus) employs a basic threshold activation mechanism that provides a prior partition boundary (activation threshold) to divide feature channels into irrelevant and relevant groups. Building upon this, intrinsic and extrinsic channel relevance assessments are derived respectively through activation-counting and decision-making processes in a data-driven manner. Finally, a posterior partition boundary between irrelevant and relevant channels is established through intrinsic-extrinsic consensus, enabling channel-wise gatekeeping that effectively suppresses irrelevant channels.

relevance. Also in Figure 1(a1), for classification tasks, each category is correlated with only a sparse and specific set of channels, indicating the presence of a significant proportion of redundant/irrelevant channels in the network, and ideally, these irrelevant channels should remain silent.

***Observation 2:*** *Linear classifier acting on activated features learns distinct decision weights for each channel, extrinsically indicating channel relevance.*

For classification tasks, we introduce a linear classifier (without the bias term) on the post-activation feature vector, whose label is the same as the main task. The learned decision weights per channel are shown in Figure 1(a2). It is observed that only a small subset of channels acquires high weights. The underlying mechanism is that: the predicted score (i.e., logit) is the weighted summation of the non-negative activation values of all channels and their corresponding weights. To increase the predicted score for the target category, the linear classifier tends to assign higher weights to important channels to amplify their impact. Consequently, the weights learned by the linear classifier can quantify the relevance of channels to the network's final decision from an extrinsic view.

***Observation 3:*** *Intrinsic and extrinsic aspects show consensus for channel relevance assessment.*

We sort Figure 1(a1) in descending order of activity level to obtain Figure 1(b1), and then re-index Figure 1(a2) following the index order of Figure 1(b1) to obtain Figure 1(b2). It can be observed that the intrinsic and extrinsic observations exhibit consistency, as they both point to some common channels, suggesting that these channels are more likely to be task-irrelevant.

## 3 METHODOLOGY

As illustrated in Figure 2, we propose AIEC (Activation with Intrinsic-Extrinsic Consensus), a novel activation mechanism designed to identify and suppress irrelevant feature channels. AIEC integrates four core components: (1) a basic Threshold Activation Unit (TAU), (2) an intrinsic Activation-Counting Unit (ACU) that tracks each channel's activation statistics and provides intrinsic channel relevance assessment, (3) an extrinsic Decision-Making Unit (DMU) that learns each channel's decision weight and provides extrinsic channel relevance assessment, and (4) a Consensus Gatekeeping Unit (CGU) that pinpoints and suppresses irrelevant channels based on the consensus of intrinsic ACU and extrinsic DMU.

### 3.1 THRESHOLD ACTIVATION UNIT

As a basic part, Threshold Activation Unit (TAU) functions as a gating mechanism for signal transmission. Similar to biological pulse neurons (Hodgkin & Huxley, 1952), given a non-negative activation threshold $\tau$, the input signal $x$ is allowed to pass to the next layer as it exceeds the threshold

$\tau$, otherwise it gets inhibited to zero. The TAU's operation on input $x$ can be expressed as

$$\text{TAU}(x) = \begin{cases} x, & if \ x > \tau \\ 0, & if \ x \le \tau \end{cases}.$$ (1)

Consider a pre-activation feature map $\mathbf{F} \in \mathbb{R}^{H \times W \times C}$ with $C$ channels. The post-activation feature map $\mathbf{A} \in \mathbb{R}^{H \times W \times C}_{[0,+\infty]}$ is obtained by applying the TAU element-wise to $\mathbf{F}$:

$$\mathbf{A} = \text{TAU}(\mathbf{F}).$$ (2)

While TAU brings some sparsity to feature representation, its effectiveness in identifying truly task-irrelevant channels and inhibiting their noise remains limited. Subsequent sections show how to overcome this by leveraging the consensus of intrinsic and extrinsic channel relevance assessment for sharper suppression.

## 3.2 INTRINSIC AND EXTRINSIC ASSESSMENTS

Operating on the post-activation feature at the channel level, this section aims to identify irrelevant channels from the intrinsic and extrinsic views, respectively.

First, perform global average pooling on the post-activation feature map $\mathbf{A}$ to obtain the global feature vector $\mathbf{a} \in \mathbb{R}^{C}_{[0,+\infty]}$:

$$\mathbf{a} = \frac{1}{HW} \sum_{h=1}^{H} \sum_{w=1}^{W} \mathbf{A}_{h,w}.$$ (3)

*For intrinsic channel relevance assessment,* Activation-Counting Unit (ACU) intrinsically quantifies channel relevance by statistically measuring each channel's activity level. For a given dataset with $K$ categories, an ACU maintains three counters for each category: $\{\boldsymbol{\theta}^k, \boldsymbol{\eta}^k, \boldsymbol{\phi}^k\}_{k=1}^{K}$. Over samples from category $k$, $\boldsymbol{\theta}^k \in \mathbb{N}^C$, $\boldsymbol{\eta}^k \in \mathbb{N}^C$, and $\boldsymbol{\phi}^k \in \mathbb{N}^C$ tracks each channel's historical activation statistics: the number of activations after TAU, the number of inhibitions after TAU, and the total sample count. Specifically, for an input data of category $k$, denote $\mathbf{a}^k \in \mathbb{R}^{C}_{[0,+\infty]}$ as its channel-level activated feature (Eq. 3). Since we only focus on whether a channel is activated or not, $\mathbf{a}^k$ is binarized to $\tilde{\mathbf{a}}^k \in \mathbb{1}^C$ as follows:

$$\tilde{\mathbf{a}}_c^k = \begin{cases} 1, & if \ \mathbf{a}_c^k > 0 \\ 0, & if \ \mathbf{a}_c^k = 0 \end{cases}.$$ (4)

Next, based on $\tilde{\mathbf{a}}^k$, the ACU updates the activation statistics $\boldsymbol{\theta}^k$, $\boldsymbol{\eta}^k$, and $\boldsymbol{\phi}^k$ as follows:

$$\boldsymbol{\theta}^{k,t} = \boldsymbol{\theta}^{k,t-1} + \tilde{\mathbf{a}}^k, \qquad \boldsymbol{\eta}^{k,t} = \boldsymbol{\eta}^{k,t-1} + (\mathbf{1} - \tilde{\mathbf{a}}^k), \qquad \boldsymbol{\phi}^{k,t} = \boldsymbol{\phi}^{k,t-1} + \mathbf{1},$$ (5)

where $t-1$ and $t$ denote previous and current iteration, respectively. $\mathbf{1} \in \mathbb{1}^C$ is a vector full of one. In implementation, only samples enabling correct predictions from the Decision-Making Unit described below are included in the statistics to obtain more accurate results. From these statistics, we define "relative firing rate" $\boldsymbol{\nu}^k \in \mathbb{R}^{C}_{[-1,1]}$, which quantifies each channel's activity level for category $k$:

$$\boldsymbol{\nu}^{k,t} = \frac{\boldsymbol{\theta}^{k,t} - \boldsymbol{\eta}^{k,t}}{\boldsymbol{\phi}^{k,t}}.$$ (6)

According to the principle of threshold activation, signals from irrelevant channels tend to be inhibited when passing through the TAU, resulting in these channels' low activity level. The intrinsic channel relevance division for each category is represented by $\{\mathcal{G}^{k,Intr}\}_{k=1}^{K}$, where $Intr$ denotes the term "Intrinsic". With zero as the natural demarcation point, $\mathcal{G}^{k,Intr} \in \mathbb{1}^C$ can be established as

$$\mathcal{G}_c^{k,Intr} = \begin{cases} 0, & if \ \boldsymbol{\nu}_c^k > 0 \\ 1, & if \ \boldsymbol{\nu}_c^k \le 0 \end{cases},$$ (7)

where the indicator "1" signifies potential irrelevant channels.

*For extrinsic channel relevance assessment,* Decision-Making Unit (DMU) extrinsically quantifies channel relevance from the natural reflection of the network's decision. A DMU contains $K$ learnable decision weights $\{\mathbf{w}^k\}_{k=1}^{K}$, respectively for the $K$ categories, where $\mathbf{w}^k \in \mathbb{R}^C$. For an input data of

category $k$, denote $\mathbf{a}^k \in \mathbb{R}_{[0,+\infty]}^C$ as its channel-level activated feature (Eq. 3). Then, linear projection and softmax are applied to $\mathbf{a}^k$ to obtain the intermediate predictions $\{p^k\}_{k=1}^K$ for each category:

$$\{p^k\}_{k=1}^K = \text{softmax}(\mathbf{a}^k \cdot \{\mathbf{w}^{k\text{T}}\}_{k=1}^K), \tag{8}$$

where $p^k \in [0, 1]$ is the predicted probability for category $k$. After that, compute the auxiliary loss aiming to minimize the difference between the prediction $p^k$ and its label $y^k \in \{0, 1\}$ as follows:

$$\mathcal{L}_{aux} = -\sum_{k=1}^K y^k \log(p^k). \tag{9}$$

The decision weight $\mathbf{w}^k \in \mathbb{R}^C$ being learned through $\mathcal{L}_{aux}$ continuously quantifies each channel's contribution/relevance to the network's decision for category $k$. The extrinsic channel relevance division for each category is represented by $\{\mathcal{G}^{k,Extr}\}_{k=1}^K$, where $Extr$ denotes the term "Extrinsic". With zero as the natural demarcation point, $\mathcal{G}^{k,Extr} \in \mathbb{1}^C$ can be established as

$$\mathcal{G}_c^{k,Extr} = \begin{cases} 0, & if \ \mathbf{w}_c^k > 0 \\ 1, & if \ \mathbf{w}_c^k \leq 0 \end{cases}, \tag{10}$$

where the indicator "1" signifies potential irrelevant channels.

### 3.3 Gatekeeping with Consensus

***With the consensus of intrinsic and extrinsic channel relevance assessment,*** Consensus Gatekeeping Unit (CGU) is performed to clean the responses from irrelevant channels during training. The final channel relevance division $\{\mathcal{G}^k\}_{k=1}^K$ for each category is the intersection (i.e., logical "and") of the intrinsic division and the extrinsic division:

$$\{\mathcal{G}^k\}_{k=1}^K = \{\mathcal{G}^{k,Intr} \cap \mathcal{G}^{k,Extr}\}_{k=1}^K. \tag{11}$$

Based on this consensus, given an input data of category $k$, first filter out irrelevant channels from $\mathbf{a}^k$ by $\mathcal{G}^k$:

$$\check{\mathbf{a}}^k = \mathbf{a}^k \odot \mathcal{G}^k, \tag{12}$$

where $\odot$ denotes the Hadamard Product. Then, impose gatekeeping on the filtered irrelevant channels by constructing a new loss item $\mathcal{L}_{gate}$ as follows:

$$\mathcal{L}_{gate} = \frac{\sum_{c=1}^C \left\| \check{\mathbf{a}}_c^k \right\|_2}{\sum_{c=1}^C \mathcal{G}_c^k}. \tag{13}$$

In this way, only the activation responses from channels with low relevance will be gradually suppressed, while the ones from other channels with high relevance are preserved.

### 3.4 Neural Network Learning

The proposed AIEC (Activation with Intrinsic-Extrinsic Consensus) can replace the network's original activation mechanisms. The gatekeeping applies to the network's activated global feature. Networks that extract feature maps/sequences compute the global feature by taking the global average of the feature maps/sequences along the channels. Regarding some Transformer models that incorporate a class token, simply peel off the class token separately as the global feature while applying the corresponding gatekeeping. Additionally, for Transformer models, the proposed AIEC is applied to each block, as Transformers excel in capturing global context throughout, while for CNN models, AIEC is applied to the last block since high-level semantics only exist in deep representations (Raghu et al., 2021). The counters in ACU are cleared at the beginning of each epoch to guarantee the timeliness and accuracy of statistics. The final loss $\mathcal{L}$ is expressed as

$$\mathcal{L} = \mathcal{L}_{task} + \lambda_{aux} \cdot \frac{1}{N} \sum_{n=1}^N \mathcal{L}_{aux}^n + \lambda_{gate} \cdot \frac{1}{N} \sum_{n=1}^N \mathcal{L}_{gate}^n, \tag{14}$$

where $\mathcal{L}_{task}$ is the primary loss for the specific task; for example, in the context of a standard classification task, $\mathcal{L}_{task}$ represents the cross-entropy loss. $N$ is the number of layers in the network that have AIEC applied, and $\lambda_{\{aux,gate\}}$ are balancing parameters.

# 4 EXPERIMENTAL STUDY

**Datasets.** We adopt seven datasets, including four vision datasets: CIFAR-10 (Krizhevsky et al., 2009), CIFAR-100 (Krizhevsky et al., 2009), ImageNet-100 (Deng et al., 2009), and ImageNet-1K (Deng et al., 2009), and three non-vision datasets: Elliptic (Weber et al., 2019), T-Finance (Tang et al., 2022), and Weibo21 (Nan et al., 2021), to verify the effectiveness of the proposed AIEC.

**Compared methods.** The proposed AIEC is compared with different types of mainstream activation mechanisms mentioned in Related Work §5.1, including Softplus (Dugas et al., 2000), ELU (Clevert et al., 2015), SELU (Klambauer et al., 2017), SiLU (Ramachandran et al., 2017), ReLU (Glorot et al., 2011), GELU (Hendrycks & Gimpel, 2016), and GDN (Ballé et al., 2015).

**Experimental settings.** The image size of CIFAR-{10,100} remains 32×32, while the images in ImageNet-{100,1K} are uniformly scaled to 224×224. To ensure the generality of the network, the activation threshold $\tau$ is uniformly set to 0. The balancing parameter $\lambda_{aux}$ for the auxiliary loss $\mathcal{L}_{aux}$ is empirically set to 1 to match the magnitude of the task loss $\mathcal{L}_{task}$, and the balancing parameter $\lambda_{gate}$ for the gatekeeping loss $\mathcal{L}_{gate}$ varies depending on networks and datasets as discussed in Appendix §A.1. All experiments use the same data augmentations provided by timm (Wightman, 2019), AdamW optimizer with weight decay of 0.05, drop-path rate of 0.1, gradient clipping norm of 1.0, and cosine annealing learning rate scheduler with linear warm-up. All experiments are trained for 300 epochs from scratch. The automatic mixed precision training strategy is adopted to speed up the training. All other training settings, including batch size, learning rate, warm-up epochs, and so on, are kept identical throughout each set of comparative experiments. Note that the numerical results are the average under three different random seeds, and no pre-training is used.

## 4.1 AIEC ON VITS AND CNNS

The proposed AIEC can be incorporated into popular Vision Transformer (ViT) and its variants. Table 1 shows the top-1 accuracy (%) across CIFAR-{10,100} and ImageNet-{100,1K} using the proposed AIEC on five different ViT architectures: ViT (Dosovitskiy et al., 2020), DeiT (Touvron et al., 2021a), CaiT (Touvron et al., 2021b), PVT (Wang et al., 2021), and TNT (Han et al., 2021). The proposed AIEC can replace all the existing activations in each block. The results consistently illustrate that AIEC outperforms the baselines, showcasing its robustness.

The proposed AIEC is also evaluated on various mainstream CNNs, including AlexNet (Krizhevsky et al., 2017), VGG (Simonyan & Zisserman, 2014), MobileNet (Howard et al., 2017), ShuffleNet(V2) (Ma et al., 2018), and ResNet (He et al., 2016). The proposed AIEC replaces the original activations in the last block since previous works have shown that high-level semantics in CNNs only exist in deep representations (Raghu et al., 2021). The results in Table 2 highlight the versatility and robustness of AIEC in handling diverse CNN architectures and datasets.

## 4.2 ABLATION STUDY

Ablation study is conducted, as presented in Table 3. First, solely introducing $\mathcal{L}_{aux}$ has no effect, indicating that the actual performance contribution of AIEC comes from $\mathcal{L}_{gate}$. Moreover, ACU including all samples for statistics does not perform as well as only counting those that lead to correct DMU predictions. Furthermore, the performance with the intrinsic-extrinsic consensus ($\text{AIEC}_{I \cap E}$) is better than using a single $\text{AIEC}_I$ or $\text{AIEC}_E$. The consensus of the two assessments ($\text{AIEC}_{I \cap E}$) is better than the combination of both ($\text{AIEC}_{I \cup E}$), as the consensus can reduce misjudgments. Finally, indiscriminately suppressing all channels ($\text{AIEC}_{all}$) yields no performance improvement.

## 4.3 COMPUTATIONAL COSTS

ACU, DMU, and CGU only work during the training phase, and in the inference phase, only TAU needs to be involved. Table 4 presents the computational costs during training and inference regarding "GPU Memory (GiB)" and "Latency (s)" (the average time it takes for a network to process a batch of data). Notably, the activation methods used in original networks should be implemented manually as our AIEC does. Using the methods directly from pre-made libraries (like torch.nn) can result in unfair comparisons due to their high optimization at the low level. Table 4 shows that AIEC's training GPU overhead is negligible and its inference speed is on par with other methods.

Table 1: Top-1 accuracy (%) across the CIFAR-10, CIFAR-100, ImageNet-100, and ImageNet-1K datasets using the proposed AIEC on Vision Transformer (ViT) and its variants.

| Top-1 Acc / % | | Softplus | ELU | SELU | SiLU | ReLU | GELU | GDN | AIEC |
|---|---|---|---|---|---|---|---|---|---|
| CIFAR-10 | ViT-Tiny | 84.3 | 82.0 | 79.4 | 85.5 | 89.9 | 89.2 | 81.8 | **91.8** |
| | DeiT-Tiny | 84.7 | 81.4 | 79.9 | 86.6 | 89.6 | 89.2 | 83.0 | **91.9** |
| | CaiT-XXS | 82.5 | 80.7 | 78.4 | 86.6 | 89.4 | 88.7 | 80.0 | **90.5** |
| | PVT-Tiny | 90.6 | 89.3 | 85.4 | 92.5 | 93.0 | 92.8 | 82.8 | **94.0** |
| | TNT-Small | 88.3 | 85.4 | 83.7 | 90.5 | 90.8 | 91.1 | 85.1 | **92.7** |
| CIFAR-100 | ViT-Tiny | 62.4 | 60.0 | 57.5 | 65.5 | 65.7 | 65.4 | 59.4 | **71.1** |
| | DeiT-Tiny | 63.4 | 60.0 | 58.3 | 67.1 | 67.0 | 67.0 | 59.8 | **71.8** |
| | CaiT-XXS | 60.4 | 59.3 | 55.8 | 63.9 | 65.8 | 65.5 | 56.2 | **70.0** |
| | PVT-Tiny | 69.5 | 69.3 | 65.7 | 70.2 | 70.9 | 70.6 | 64.4 | **76.0** |
| | TNT-Small | 65.2 | 63.8 | 60.9 | 65.1 | 65.4 | 64.4 | 62.5 | **73.9** |
| ImageNet-100 | ViT-Tiny | 74.1 | 68.9 | 66.4 | 74.1 | 75.4 | 76.4 | 67.9 | **82.1** |
| | DeiT-Tiny | 75.3 | 69.4 | 67.0 | 75.1 | 75.6 | 74.6 | 66.3 | **82.5** |
| | CaiT-XXS | 70.9 | 69.1 | 65.9 | 76.1 | 76.0 | 76.7 | 69.5 | **81.3** |
| | PVT-Tiny | 79.5 | 77.1 | 76.1 | 79.5 | 81.9 | 81.4 | 75.8 | **86.3** |
| | TNT-Small | 78.9 | 79.3 | 76.4 | 77.6 | 79.9 | 77.2 | 76.9 | **86.5** |
| ImageNet-1K | ViT-Tiny | 70.0 | 64.2 | 63.1 | 66.9 | 70.9 | 70.4 | 65.2 | **73.0** |
| | DeiT-Tiny | 71.9 | 67.9 | 66.2 | 72.0 | 73.2 | 73.0 | 66.4 | **73.7** |
| | CaiT-XXS | 70.3 | 68.1 | 66.7 | 73.2 | 74.0 | 73.6 | 66.1 | **74.1** |
| | PVT-Tiny | 71.5 | 69.2 | 68.5 | 72.8 | 73.7 | 73.5 | 66.5 | **74.6** |
| | TNT-Small | 72.0 | 70.7 | 70.3 | 71.5 | 73.4 | 73.3 | 68.2 | **78.1** |

Table 2: Top-1 accuracy (%) across the CIFAR-10, CIFAR-100, ImageNet-100, and ImageNet-1K datasets using the proposed AIEC on various CNN architectures.

| Top-1 Acc / % | | Softplus | ELU | SELU | SiLU | ReLU | GELU | GDN | AIEC |
|---|---|---|---|---|---|---|---|---|---|
| CIFAR-10 | AlexNet | 85.6 | 86.1 | 85.7 | 86.0 | 86.0 | 85.8 | 85.4 | **86.3** |
| | VGG-11 | 91.3 | 92.0 | 91.5 | 91.9 | **92.2** | 91.9 | 91.1 | **92.2** |
| | MobileNet | 87.4 | 87.7 | 87.2 | 87.8 | 87.4 | 87.4 | 87.0 | **89.0** |
| | ShuffleNet | 89.2 | 89.0 | 88.9 | 89.3 | 89.4 | 89.3 | 88.5 | **89.9** |
| | ResNet-18 | 94.6 | 94.7 | 94.6 | **95.1** | 95.0 | 94.9 | 94.0 | **95.1** |
| CIFAR-100 | AlexNet | 57.6 | 58.4 | 58.1 | 58.1 | 57.2 | 57.4 | 56.8 | **58.9** |
| | VGG-11 | 69.6 | 69.9 | 69.7 | 69.9 | 70.2 | 70.0 | 70.1 | **71.2** |
| | MobileNet | 65.4 | 65.5 | 65.6 | 65.2 | 66.0 | 65.4 | 64.8 | **66.1** |
| | ShuffleNet | 66.2 | 66.1 | 65.9 | 66.3 | 66.3 | 66.2 | 65.6 | **66.9** |
| | ResNet-18 | 75.5 | 75.7 | 75.6 | 76.1 | 75.7 | 75.6 | 74.3 | **77.0** |
| ImageNet-100 | AlexNet | 75.7 | 76.0 | 75.7 | 76.6 | 76.3 | 76.3 | 75.5 | **79.2** |
| | VGG-11 | 87.0 | 87.3 | 87.6 | 87.8 | 87.7 | 87.5 | 86.7 | **88.1** |
| | MobileNet | 80.6 | 79.3 | 79.2 | 80.1 | 80.6 | 80.5 | 78.7 | **80.8** |
| | ShuffleNet | 80.9 | 80.9 | 80.4 | 81.7 | 81.6 | 81.6 | 80.0 | **82.2** |
| | ResNet-18 | 84.6 | 84.4 | 84.1 | 84.9 | 84.9 | 84.7 | 83.5 | **86.5** |
| ImageNet-1K | AlexNet | 56.1 | 56.3 | 56.1 | 56.4 | 56.5 | 56.4 | 55.6 | **57.8** |
| | VGG-11 | 68.4 | 68.2 | 67.8 | 69.0 | 69.0 | 69.1 | 68.1 | **70.1** |
| | MobileNet | 67.2 | 66.7 | 67.1 | 67.4 | 68.1 | 68.2 | 66.3 | **68.6** |
| | ShuffleNet | 68.5 | 68.3 | 68.4 | 69.1 | 69.0 | 68.9 | 68.0 | **69.6** |
| | ResNet-18 | 69.3 | 69.4 | 68.9 | 69.7 | 69.7 | 69.4 | 68.3 | **70.4** |

Table 3: Ablation study on CIFAR-100. AIEC$_{\mathcal{L}_{aux}}$ denotes using the auxiliary loss $\mathcal{L}_{aux}$ only. AIEC$_{stats\text{-}all}$ denotes including all samples for ACU's statistics. AIEC$_I$, AIEC$_E$, AIEC$_{I \cup E}$, and AIEC$_{I \cap E}$ denote AIEC with the channel relevance assessment in the form of intrinsic only, extrinsic only, intrinsic-extrinsic combination, and intrinsic-extrinsic consensus, respectively. AIEC$_{all}$ denotes suppressing all channels indiscriminately.

| Top-1 Acc / % | AIEC$_{\mathcal{L}_{aux}}$ | AIEC$_{stats\text{-}all}$ | AIEC$_{all}$ | AIEC$_I$ | AIEC$_E$ | AIEC$_{I \cup E}$ | AIEC$_{I \cap E}$ |
|---|---|---|---|---|---|---|---|
| ViT-Tiny | 66.7 | 70.3 | 65.8 | 69.7 | 68.5 | 69.6 | **71.1** |
| ResNet-18 | 76.1 | 76.8 | 75.7 | 76.3 | 76.6 | 76.6 | **77.0** |

Table 4: Computational costs during training and inference regarding "GPU Memory (GiB)" and "Latency (s)". The networks were fed 224×224-pixel images with a batch size of 1024 on an NVIDIA A6000 GPU. "Latency" refers to the average time it takes for a network to process a batch of data.

| Computational Costs | ViT-Tiny | | | ResNet-18 | | |
|---|---|---|---|---|---|---|
| | ReLU | GELU | AIEC | ReLU | GELU | AIEC |
| Training GPU Memory / GiB | 32.2 | 28.3 | 32.5 | 24.3 | 27.9 | 24.8 |
| Inference GPU Memory / GiB | 4.8 | 4.8 | 4.9 | 7.4 | 7.4 | 7.4 |
| Training Latency / ms | 540.7 | 640.9 | 592.0 | 417.5 | 437.6 | 421.9 |
| Inference Latency / ms | 9.2 | 10.2 | 9.2 | 7.5 | 8.1 | 7.5 |

Table 5: The generalization performance of the proposed AIEC on the non-vision anomaly detection tasks including finance fraud detection and fake news detection.

| Dataset | T-Finance | | | Elliptic | | | Weibo | | |
|---|---|---|---|---|---|---|---|---|---|
| Metric | AUC | AP | Rec@K | AUC | AP | Rec@K | AUC | AP | Rec@K |
| GCN-ReLU | 92.9 | 75.8 | 70.6 | **81.1** | 21.3 | 25.4 | 98.5 | 94.0 | **90.2** |
| GCN-GELU | 93.0 | 76.2 | 72.8 | 77.4 | 15.2 | 10.1 | **98.6** | 94.0 | 89.9 |
| GCN-AIEC | **93.1** | **77.8** | **73.4** | 80.9 | **34.9** | **42.2** | **98.6** | **95.1** | **90.2** |
| GraphSAGE-ReLU | 84.5 | 60.3 | 63.7 | 84.9 | 35.0 | 38.2 | 96.6 | 92.2 | 88.2 |
| GraphSAGE-GELU | 84.9 | 52.9 | 58.5 | 85.0 | 35.1 | 41.0 | 97.3 | 93.3 | **89.3** |
| GraphSAGE-AIEC | **90.9** | **70.0** | 67.3 | **85.6** | **43.1** | **46.1** | **98.3** | **93.4** | **89.3** |

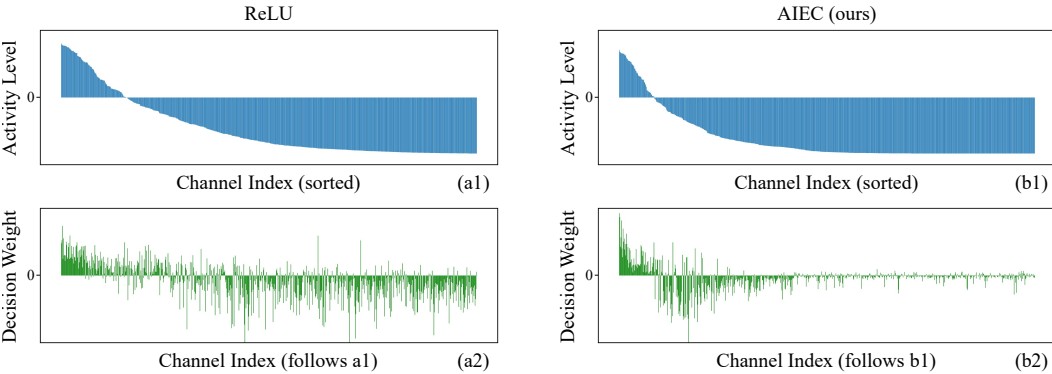

Figure 3: Visualization results. The intrinsic activity level (a1, b1) and extrinsic decision weight (a2, b2) are recorded for each channel of the feature vector after the activation (ReLU vs AIEC) in the last block of ViT-Tiny, specifically for the "truck" category in the CIFAR-10 dataset. More results are provided in Figure 6, 7, 8.

## 4.4 GENERALIZATION TO OTHER TASKS

The proposed AIEC can also perform various other tasks or domains, like the non-vision task anomaly detection, including finance fraud detection and fake news detection. For finance fraud detection, the Elliptic (Weber et al., 2019) and T-Finance (Tang et al., 2022) datasets are employed. For fake news detection, the Weibo21 (Nan et al., 2021) dataset is employed. GCN (Kipf & Welling, 2016) and GraphSAGE (Hamilton et al., 2017) are chosen networks. The results are shown in Table 5.

## 4.5 VISUALIZATION RESULTS

As shown in Figure 3, with the proposed AIEC, the activation responses become sparser, as can be seen from the activity levels of the channels, implying that some irrelevant channels are identified and suppressed. Moreover, the channel gatekeeping in AIEC leads to clearer sign boundaries in the decision weights, reflecting that the network becomes more confident in assigning importance to channels that consistently contribute to the correct decision. These phenomena suggest that AIEC improves the precision of key feature extraction, which reduces learning difficulty and increases interpretability. More visualizations and discussions are provided in Appendix §A.4.

## 5 RELATED WORK

### 5.1 ACTIVATION MECHANISMS

The activation mechanism (Dubey et al., 2022) plays a pivotal role in ANNs as it defines how neurons respond to input signals, convert them into output signals, and transmit them to the subsequent layer. Activation mechanisms are categorized into types, including logistic Sigmoid and Tanh variants (LeCun et al., 1998), Rectified Linear Unit variants (Glorot et al., 2011), Exponential Linear Unit variants (Clevert et al., 2015), Softplus variants (Dugas et al., 2000), probabilistic variants (Hendrycks & Gimpel, 2016), and others (Ramachandran et al., 2017; Ballé et al., 2015; Lee et al., 2022).

In the widely used form of activation (LeCun et al., 1998; Glorot et al., 2011; Dugas et al., 2000; Clevert et al., 2015; Hendrycks & Gimpel, 2016), irrelevant features are inhibited, and relevant features gain amplified influence according to the response rule of the neuron. Furthermore, some activation mechanisms (LeCun et al., 1998; Glorot et al., 2011; Dugas et al., 2000) can effectively achieve data sparsity, diminish redundant information, and enable better feature distinction. Additionally, the activation mechanisms, such as ELU (Clevert et al., 2015) and SiLU (Ramachandran et al., 2017) mentioned above, and others (Liu et al., 2020), contain learnable parameters inside the activation itself. Parameters in Liu et al. (2020) can adapt to various data distributions, avoiding gradient vanishing and explosion, thereby enhancing the convergence speed and precision of ANNs. However, these extra parameters can only uniformly influence the response strength for all inputs, and are learned without explicit supervision. Consequently, like ordinary activation mechanisms, they primarily rely on instantaneous activation intensity to gate feature propagation, lacking interpretable evidence for channel relevance awareness.

### 5.2 FEATURE ATTRIBUTION TECHNIQUES

Feature attribution (Mandler & Weigand, 2024) aims to assess each feature's contribution to the network's final decision, distinguishing relevant from irrelevant features. Class Activation Mapping (CAM) (Zhou et al., 2016) computes the saliency map by projecting back the weights of the output layer onto the input feature maps, which motivates us to explore similar patterns and design extrinsic assessment for channel relevance in our study. Grad-CAM (Selvaraju et al., 2017) extends this to any layer via the gradient relative to the target class. Built on it, Model Doctor (Feng et al., 2022) applies constraints on erroneous channel gradients to correct the decision stream. GradToken (Cheng et al., 2025) exploits class-aware gradients to decouple the tangled semantics in the class token and leverages class-spatial token relations to generate relevance maps. Layer-wise Relevance Propagation (LRP) (Bach et al., 2015) decomposes decisions by propagating attributions backward through the network. Chefer et al. (2021), Wu & Ong (2021), and Ali et al. (2022) further extend and apply the LRP technique in various tasks. Perturbation-based methods (Ivanovs et al., 2021) operate by manipulating input pixels and observing output changes. Geva et al. (2022), Deiseroth et al. (2023), and Vilas et al. (2023) project internal representations into a human-understandable class embedding space, and then determine the relevance of different image regions to the target class. FovEx (Panda et al., 2025) combines biologically inspired foveation-based transformations with gradient-driven overt attention to iteratively assess the relevance of each location.

These techniques inspire us to design a novel activation mechanism with an awareness of channel relevance, based on each channel's contribution to the network's decision, and is able to effectively suppress interference from irrelevant channels.

## 6 CONCLUSION

In this paper, we propose AIEC (Activation with Intrinsic-Extrinsic Consensus), a novel activation mechanism that identifies and suppresses irrelevant feature channels during training by leveraging the consensus between intrinsic activity levels and extrinsic decision weights. Through extensive experiments across vision and non-vision tasks, we demonstrate that AIEC consistently outperforms existing activation mechanisms, promotes sparser and more interpretable representations, and generalizes effectively across diverse architectures, all with minimal computational overhead. Our work highlights the importance of explicit channel relevance assessment and represents a step toward more robust and effective activation mechanisms in deep learning.

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

# A APPENDIX

## A.1 HYPERPARAMETER IMPACT ANALYSIS

**Activation threshold $\tau$ for the Threshold Activation Unit (TAU).** To achieve preliminary sparsity, only input signals that exceed the threshold $\tau$ are activated, while those below the threshold are inhibited to zero. Therefore, $\tau$ is a non-negative value. Figure 4 demonstrates that as $\tau$ increases, the performance decreases. Possible reasons could be the influence of weight initialization and feature normalization operations. Typically, weights are initialized using a distribution with a mean of zero, and normalization techniques such as layer normalization and batch normalization are used to make the feature distribution centered around zero (by subtracting the feature mean) to eliminate shifts in data covariates. Under these circumstances, $\tau = 0$ becomes the optimal activation threshold. Additionally, it may be feasible to modify the strategy for weight initialization and feature normalization to achieve optimal effects when considering a positive $\tau$.

**Balancing parameter $\lambda_{gate}$ for the gatekeeping loss $\mathcal{L}_{gate}$.** The optimal $\lambda_{gate}$ for $\mathcal{L}_{gate}$ is specific to individual tasks. The relationship between $\lambda_{gate}$ and the accuracy on CIFAR-100 with ViT-Tiny is depicted in Figure 5. In this case, the optimal $\lambda_{gate}$ is roughly 200, and too large a $\lambda_{gate}$ can result in negative side effects. For other trials, we found the optimal $\lambda_{gate}$ to be 400 and 200 respectively when training DeiT-Tiny and TNT-Small on CIFAR-100 and the optimal $\lambda_{gate}$ to be 200 when training ViT-Tiny on CIFAR-10. Searching for the optimal $\lambda_{gate}$ for each task is time-consuming. Therefore, for the majority of our experiments, we set the default $\lambda_{gate}$ to 200.

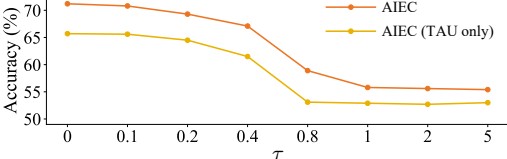 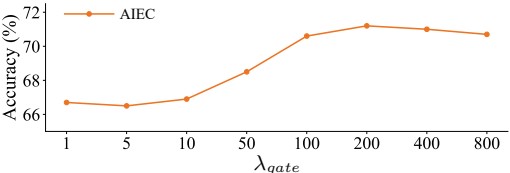

Figure 4: Top-1 accuracy (%) w.r.t. the activation threshold $\tau$ for the Threshold Activation Unit (TAU) when training on CIFAR-100 with ViT-Tiny.

Figure 5: Top-1 accuracy (%) w.r.t. the balancing parameter $\lambda_{gate}$ for $\mathcal{L}_{gate}$ when training on CIFAR-100 with ViT-Tiny.

## A.2 DIFFERENCES BETWEEN CHANNEL GATEKEEPING AND CHANNEL PRUNING

The proposed channel gatekeeping differs from channel pruning as follows:

- The target of channel pruning is channel weights $\mathbf{W}$, while the target of channel gatekeeping is channel responses $f(\text{input}; \mathbf{W})$. Channel pruning operates by setting **the weights of certain channels** to zero, resulting in no responses on pruned channels for any input (This is why channel pruning typically leads to accuracy decrease.). In contrast, channel gatekeeping operates by suppressing **the responses of certain channels** to zero, in which case, the channel weights are not necessarily zero, and the suppressed channels can vary for different inputs.

- Channel pruning requires post-processing to remove irrelevant channel weights and some need further fine-tuning, while channel gatekeeping is conveniently trained end-to-end from scratch and does not require any post-processing or fine-tuning.

- The objective of channel pruning is trying to reduce computation and storage requirements without sacrificing accuracy, while the objective of channel gatekeeping is trying to improve accuracy without increasing computational overhead.

## A.3 POTENTIAL IN REAL-WORLD APPLICATIONS

The proposed AIEC has been designed to be adaptable to various ANN architectures including CNNs, ViTs, and GNNs with minimal computational overhead. The code is modularized and can be easily integrated into existing systems without requiring extra special hardware support. The internal mechanism of AIEC can accommodate data from various categories and domains. The work performed has no negative societal impact.

## A.4 MORE VISUALIZATION RESULTS

This section extends §4.5 of the main paper, providing visualizations for all categories in CIFAR-10, as shown in Figure 6, 7, 8. With the proposed AIEC, the activation responses become sparser, as evident from the activity levels across channels. This implies that AIEC has identified and suppressed some irrelevant channels. Furthermore, channel gatekeeping in the AIEC results in sharper decision weight boundaries, demonstrating that the network gains greater confidence in prioritizing channels that consistently support correct decisions. These suggest that the AIEC enhances the precision of key feature extraction, reducing learning difficulty and improving model interpretability. Notably, more than half of the channels exhibit their decision weights close to zero, which corresponds to their activity levels dropping to a minimum, meaning these channels are completely inhibited. As a result, the decision weights learn no useful information from them. In contrast, the other active channels provide meaningful input to the decision weights, allowing the weights to assess each channel's contribution/relevance and reflect it in the sign and magnitude of the weights.

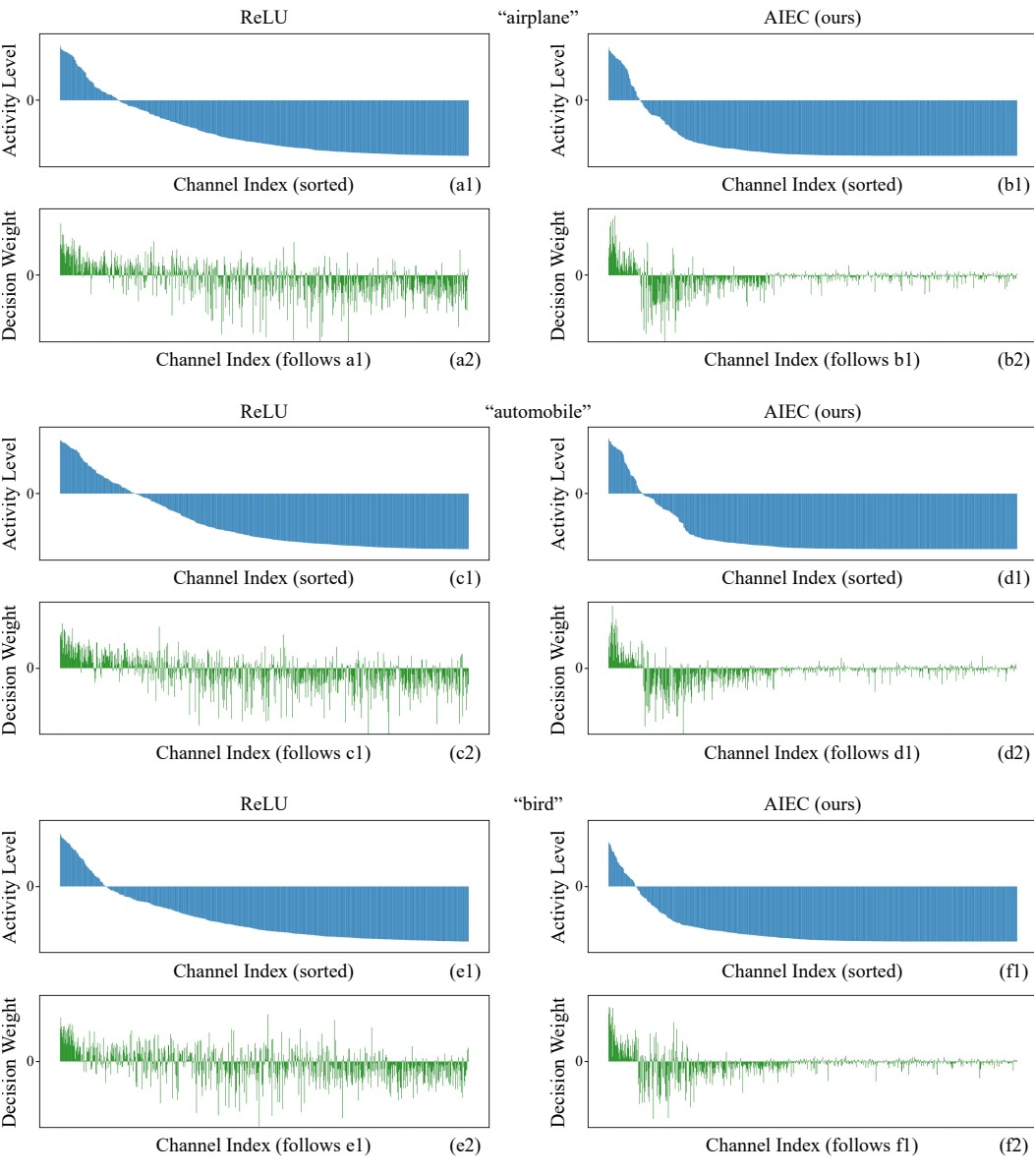

Figure 6: More visualization results extending Figure 3 of the main paper, specifically for the "airplane", "automobile", and "bird" categories in the CIFAR-10 dataset.

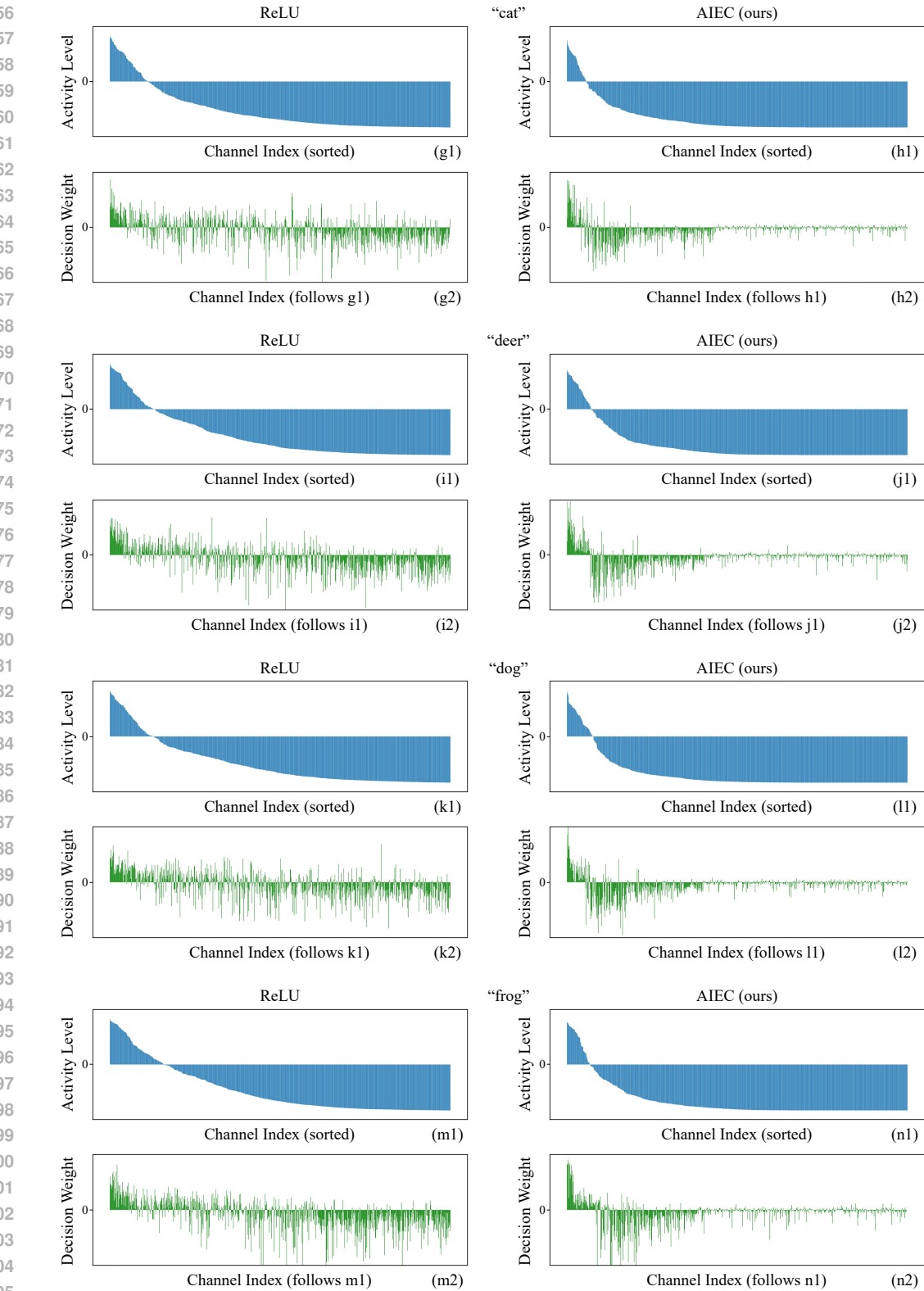

Figure 7: More visualization results extending Figure 3 of the main paper, specifically for the "cat", "deer", "dog", and "frog" categories in the CIFAR-10 dataset.

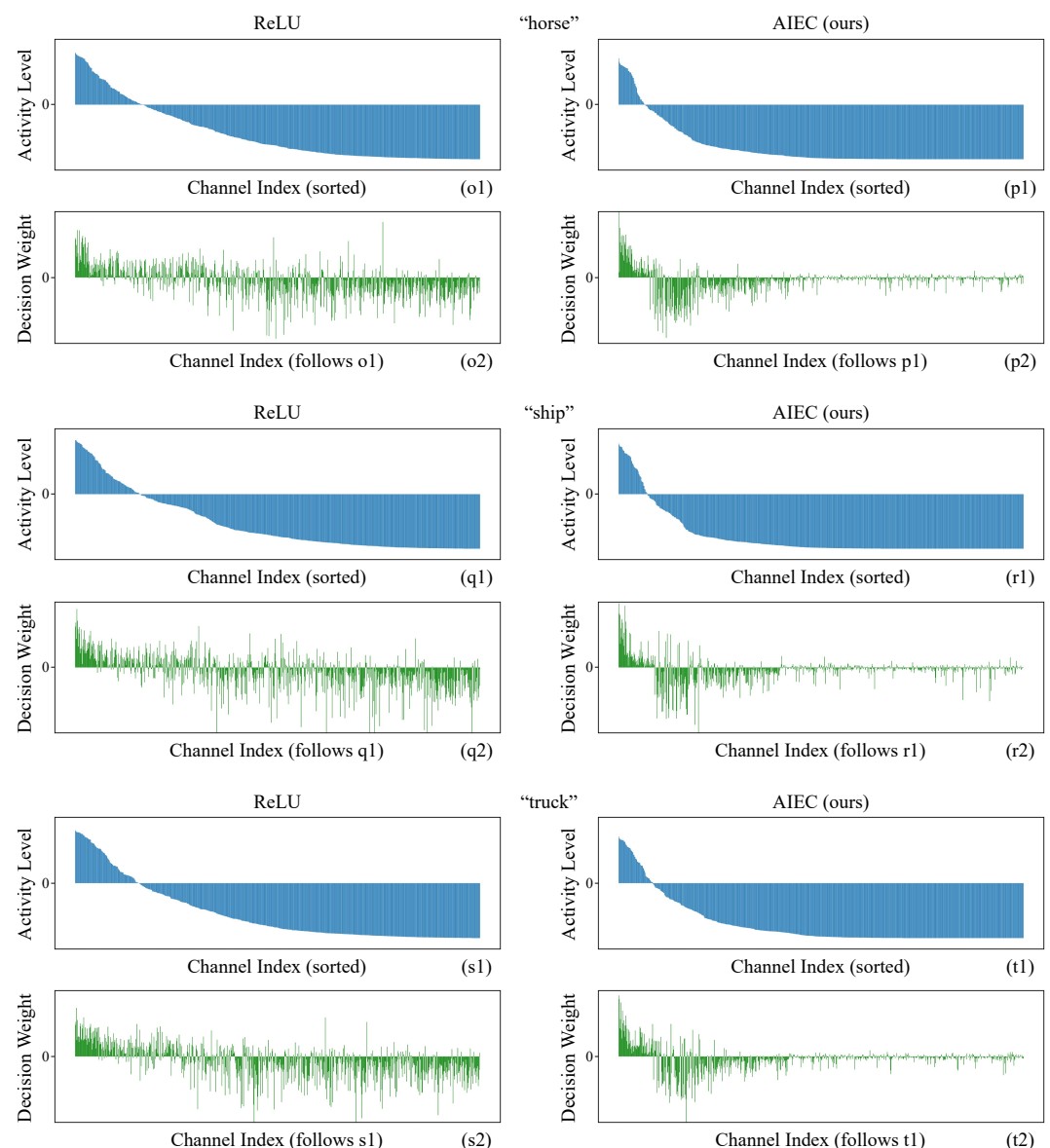

Figure 8: More visualization results extending Figure 3 of the main paper, specifically for the "horse", "ship", and "truck" categories in the CIFAR-10 dataset.

## A.5 TRAINING AND INFERENCE PROCEDURES

This section provides a comprehensive description of the training and inference procedures as elucidated in Algorithm 1 and 2. During the training phase, TAU, ACU, DMU, and CGU work together. The responses from irrelevant channels are suppressed under the guidance of the gatekeeping loss $\mathcal{L}_{gate}$ (Eq. 13). By the end of training, the model has already learned how to avoid generating responses on irrelevant channels. Hence, during the inference phase, ACU, DMU, and CGU are all discarded, and only TAU needs to be involved.

## A.6 DETAILED EXPERIMENTAL SETUP

This section provides a more detailed experimental setup including model settings and experimental settings used in this study, as listed in Table 6 and 7.

---

**Algorithm 1** : Training procedure

---

**Input:** $\mathbb{D}_{train}^K$: Training dataset with $K$ categories; $T$: Maximum number of iterations; $l$: Layer index where AIEC is applied (one layer as an example here; multiple layers are possible).

**Initialize:** $f_{1:L}(\cdot)$: Model with $L$ layers; $\{\theta^k, \eta^k, \phi^k\}_{k=1}^K$: ACU's counters; $\{\mathbf{w}^k\}_{k=1}^K$: DMU's decision weights; $\lambda_{aux}, \lambda_{gate}$: Balacing parameters.

1: **for** $t = 1, 2, ..., T$ **do**
2:      Sample $\{\mathbf{X}^k, y^k\}$ from $\mathbb{D}_{train}^K$, which is of category $k$.
3:      Compute pre-activation feature map $\mathbf{F}^k = f_{1:l}(\mathbf{X}^k)$.
4:      Compute post-activation feature map $\mathbf{A}^k = \mathrm{TAU}(\mathbf{F}^k)$.

5:      Compute post-activation feature vector $\mathbf{a}^k = \frac{1}{HW} \sum_{h=1}^H \sum_{w=1}^W \mathbf{A}_{h,w}^k$.

6:      Binarize $\mathbf{a}^k$ to $\tilde{\mathbf{a}}_c^k = \begin{cases} 1, & if \;\; \mathbf{a}_c^k > 0 \\ 0, & if \;\; \mathbf{a}_c^k = 0 \end{cases}$ based on whether a channel is activated or not.
7:      Update counters $\theta^{k,t} = \theta^{k,t-1} + \tilde{\mathbf{a}}^k, \;\; \eta^{k,t} = \eta^{k,t-1} + (\mathbf{1} - \tilde{\mathbf{a}}^k), \;\; \phi^{k,t} = \phi^{k,t-1} + \mathbf{1}$.
8:      Compute "relative firing rate" $\mathbf{v}^{k,t} = \frac{\theta^{k,t} - \eta^{k,t}}{\phi^{k,t}}$.
9:      Establish intrinsic channel relevance division $\mathcal{G}_c^{k,Intr} = \begin{cases} 0, & if \;\; \mathbf{v}_c^k > 0 \\ 1, & if \;\; \mathbf{v}_c^k \leq 0 \end{cases}$.

10:     Obtain intermediate predictions $\{p^k\}_{k=1}^K = \mathrm{softmax}(\mathbf{a}^k \cdot \{\mathbf{w}^{k\mathrm{T}}\}_{k=1}^K)$.
11:     Compute auxiliary loss $\mathcal{L}_{aux} = -\sum_{k=1}^K y^k \log(p^k)$ for learning $\{\mathbf{w}^k\}_{k=1}^K$.
12:     Establish extrinsic channel relevance division $\mathcal{G}_c^{k,Extr} = \begin{cases} 0, & if \;\; \mathbf{w}_c^k > 0 \\ 1, & if \;\; \mathbf{w}_c^k \leq 0 \end{cases}$.

13:     Establish final channel relevance division $\{\mathcal{G}^k\}_{k=1}^K = \{\mathcal{G}^{k,Intr} \cap \mathcal{G}^{k,Extr}\}_{k=1}^K$ based on intrinsic-extrinsic consensus, where the indicator "1" signifies potential irrelevant channels.
14:     Filter out irrelevant channels $\check{\mathbf{a}}^k = \mathbf{a}^k \odot \mathcal{G}^k$.
15:     Impose gatekeeping on irrelevant channels by loss $\mathcal{L}_{gate} = \frac{\sum_{c=1}^C \|\check{\mathbf{a}}_c^k\|_2}{\sum_{c=1}^C \mathcal{G}_c^k}$.

16:     Keep passing $\mathbf{A}^k$ forward until the final layer outputs $\mathbf{logits} = f_{l+1:L}(\mathbf{A}^k)$.
17:     Compute task-level loss $\mathcal{L}_{task} = \mathrm{CrossEntropy}(\mathbf{logits}, y^k)$.
18:     Compute the final loss $\mathcal{L} = \mathcal{L}_{task} + \lambda_{aux} \cdot \mathcal{L}_{aux} + \lambda_{gate} \cdot \mathcal{L}_{gate}$.

19:     Update model parameters through back-propagation and gradient descent.
20: **end for**

---

---

**Algorithm 2** : Inference procedure

---

**Input:** $\mathbf{X}$: a test sample with its category unknown.

**Initialize:** $f_{1:L}(\cdot)$: Trained model with $L$ layers.

1: Compute pre-activation feature map $\mathbf{F} = f_{1:l}(\mathbf{X})$.
2: Compute post-activation feature map $\mathbf{A} = \mathrm{TAU}(\mathbf{F})$.
3: Keep passing $\mathbf{A}$ forward until the final layer outputs $\mathbf{logits} = f_{l+1:L}(\mathbf{A})$.
4: Predict $\mathbf{X}$'s label $\hat{y} = \mathrm{argmax}(\mathbf{logits})$.

---

Table 6: Detailed model settings. "(left) / (right)" is for training CIFAR-{10,100} (left) and ImageNet-{100,1K} (right), respectively.

| Model / Setting | patch size | embed dim | # layers | # heads | # params |
|---|---|---|---|---|---|
| ViT-Tiny | 4 / 16 | 192 | 12 | 3 | 5 M |
| DeiT-Tiny | 4 / 16 | 192 | 12 | 3 | 5 M |
| CaiT-XXS | 4 / 16 | 192 | 24 | 4 | 11 M |
| PVT-Tiny | 4 | [64, 128, 320, 512] | [2, 2, 2, 2] | [1, 2, 5, 8] | 13 M |
| TNT-Small | 4 / 16 | 384 | 12 | 6 (out) + 4 (in) | 23 M |
| AlexNet | - | - | - | - | 23 M / 61 M |
| VGG-11 | - | - | - | - | 28 M / 132 M |
| MobileNet | - | - | - | - | 3 M / 5 M |
| ShuffleNet | - | - | - | - | 1 M / 2 M |
| ResNet-18 | - | - | - | - | 11 M / 11 M |

Table 7: Detailed experimental settings. "(left) / (right)" is for training CIFAR-{10,100} (left) and ImageNet-{100,1K} (right), respectively.

| Argument | Value |
|---|---|
| epochs | 300 |
| batch size | identical in each comparison group |
| optimizer | AdamW |
| learning rate | 0.0005 × (batch_size / 512) |
| learning rate scheduler | cosine annealing |
| weight decay | 5e-2 |
| warmup epochs | 5 / 20 |
| warmup learning rate | 1e-6 |
| warmup scheduler | linear |
| dropout rate | 0.0 |
| drop path rate | 0.1 |
| gradient clipping norm | 1.0 |
| image size | 224×224 / 32×32 |
| horizontal flipping rate | 0.5 |
| vertical flipping rate | 0.0 |
| color jitter | 0.4 |
| auto augment | rand-m9-mstd0.5-inc1 (from "timm") |
| interpolation | random |
| random erasing prob | 0.25 |
| random erasing mode | pixel |
| random erasing count | 1 |
| training from scratch (no pre-training) | ✓ |
| automatic mixed precision training | ✓ |
| $\tau$ | 0 |
| $\mathcal{L}_{aux}$ | 1 |
| $\mathcal{L}_{gate}$ | depending on networks and datasets |

## A.7 COMPARISON WITH OTHER METHODS

In addition, we discuss and compare the proposed AIEC with other methods including:

$L_0$ **regularization**: Theoretically, $L_0$ regularization (Louizos et al., 2017) can constrain the number of non-zero elements in features by adding a loss term. However, when the number of features is large, such a constraint makes it difficult for the model to determine which specific elements should be suppressed to zero. In contrast, the proposed AIEC provides clear criteria indicating which elements should be suppressed, offering more sufficient supervision signals and stronger interpretability.

**Attention-based gating modules:** Modules like SENet (Hu et al., 2018) and CBAM (Woo et al., 2018) introduce additional learnable fully connected or convolutional layers to generate attention weights at the channel level (or the spatial level). It simply weights the channels without evidence

of sparse representation or interpretability. In contrast, the proposed AIEC leverages both intrinsic and extrinsic characteristics of threshold activation as direct supervision signals to identify irrelevant channels and apply noise cleaning targeted on them, thus providing strong sparsity and interpretability.

The comparison of these methods is presented in Table 8. Note that CBAM is not applicable to ViT-Tiny because it contains convolutional layers, which are incompatible with the architecture of ViT-Tiny. The results demonstrate that the proposed AIEC achieves superior performance compared to these methods.

Table 8: Comparison with other methods on the CIFAR-100 dataset.

| Top-1 Acc / % | ReLU | GELU | ReLU + $L_0$ | GELU + $L_0$ | SENet | CBAM | AIEC |
|---|---|---|---|---|---|---|---|
| ViT-Tiny | 65.7 | 65.4 | 67.1 | 67.2 | 65.7 | N/A | **71.1** |
| ResNet-18 | 75.7 | 75.6 | 76.5 | 76.2 | 75.6 | 76.2 | **77.0** |

## A.8 DECLARATION OF LLM USAGE

Large language models (LLMs) were used to polish the writing at the word (e.g., grammar, spelling, word choice) and sentence levels to enhance overall fluency and coherence.

