# OpenReview forum: "Activation with Intrinsic-Extrinsic Consensus"
_ICLR.cc/2026/Conference — Submitted to ICLR 2026_

### Official Review · Reviewer_g8BT · 2025-10-15

**Soundness:** 2
**Presentation:** 2
**Contribution:** 2
**Rating:** 2
**Confidence:** 3

**Summary:**

The idea of this paper is that some features learned by a network are not relevant and should be suppressed.  The paper proposes a mechanism to identify and then suppress features by adding some parameters and a moving average of activity for a given feature detector.  The paper proposed some additional loss functions for training these auxiliary parameters, many of which are only needed during training.  The evaluations show improved results on image classification benchmarks for ResNet-18, DeiT-Tiny, etc.

**Strengths:**

The paper provides intuitive justification for the proposed technique.

**Weaknesses:**

The idea of suppressing irrelevant features seems like something gradient descent would just want to do by default with any reasonable loss function.  Invoking a simple biological model from the 1950's is entertaining in this context, but given the reasoning is very informal about how this applies it's unclear to me how sound the reasoning is that leads to the AIEC technique.

The paper lack a comparison with models extended to increase parameter counts by the same amount as required to implementing the proposed AIEC technique.

Writing could be improved:  The phrase "decision weight" is never defined but it's used extensively including in the abstract.  Text is duplicated unnecessarily (the caption for Figure 1 is same as text on lines 149-156).  Figure 2 is referenced on Page 1 long before Figure 1 which isn't referenced until Page 3.  There is no clear separation of training versus inference in the algorithm description (indeed, there are no algorithm figures at all -- just equations).  Some equations seem incorrect.

**Questions:**

What is a "decision weight"?   The paper is written seemingly assuming readers are familiar with this term, but I haven't come across it that I can remember.

What if $a^k_c < 0$ in Equation 4?  Is there some reason to believe that Equation 3 should result in non-negative elements of $a$?

Does Equation 13 not suppress the channels with high relevance?  If the point of Equation 11 is to find channels of high relevance, this seems to me to be what would happen with the loss function in Equation 13.

Can you outline the steps required to apply your proposal in training and then in inference?  I did't see an algorithm figure in the main text or the appendices.

---

> ### Author Response · Authors · 2025-11-20
> **Author Response to Reviewer g8BT (Part 1/3)**
>
> We express our appreciation for the reviewer's constructive and helpful comments, which have significantly helped us in improving the quality of our manuscript. Additionally, we have provided detailed explanations to address certain misunderstandings and clarify potential points of confusion.
>
> &nbsp;
>
> > **W1: (a)** The idea of suppressing irrelevant features seems like something gradient descent would just want to do by default with any reasonable loss function. **(b)** Invoking a simple biological model from the 1950's is entertaining in this context, but given the reasoning is very informal about how this applies it's unclear to me how sound the reasoning is that leads to the AIEC technique.
>
> Many thanks for your thoughtful and valuable feedback.
>
> **(a)** We appreciate and agree with this viewpoint. However, while we acknowledge that gradient descent with any reasonable loss function (such as cross-entropy loss in classification tasks) inherently suppresses irrelevant features by default, **we argue that this suppression is often insufficient**. Gradient descent is a global optimization process that may **converge to local optima**, leaving some irrelevant channels inadequately penalized. As a result, noise from these channels can still propagate. Our goal is to open the "black box" of deep neural networks and **introduce more targeted and more interpretable mechanisms to guide gradient descent**, thereby achieving **better** suppression of irrelevant features.
>
> **(b)** Since artificial neural networks were originally inspired by biological neural networks, the reference to a biological model was intended to illustrate the general concept of "threshold firing". The goal of our proposed AIEC is to partition channel features into irrelevant and relevant groups. Introducing a **prior** partition boundary (activation threshold) in Threshold Activation Unit (TAU) not only preliminarily achieves this goal but also aligns with biological principles. Building on this, we derive a **posterior** partition boundary between irrelevant and relevant features during training, guided by three data-driven observations (intrinsic, extrinsic, and consensus).
>
> We acknowledge that the use of the term "biological model" may have been influenced by a "mindset" formed from our previous biologically inspired studies, which could distract readers from the core contributions. We will remove this phrasing in the revised manuscript.
>
> &nbsp;
>
> > **W2:** The paper lack a comparison with models extended to increase parameter counts by the same amount as required to implementing the proposed AIEC technique.
>
> Your constructive feedback is much appreciated.
>
> First, our method introduces an auxiliary loss $\mathcal{L}\_{aux}$ with additional linear parameters, and designs a gatekeeping loss $\mathcal{L}\_{gate}$ with no parameters. In Table 3 of our original submission, we compared (i) the proposed AIEC ($\mathcal{L}\_{aux}$ + $\mathcal{L}\_{gate}$) with (ii) the method using only $\mathcal{L}\_{aux}$. The methods (i) and (ii) both have exactly the same number of parameters. The results on CIFAR-100 are as follows:
>
> |Method|Top-1 Acc / %|
> |-|-|
> |ViT-Tiny|65.7|
> |ViT-Tiny + $\mathcal{L}\_{aux}$ (linear parameters)|66.7|
> |ViT-Tiny + $\mathcal{L}\_{aux}$ (linear parameters) + $\mathcal{L}\_{gate}$ (no parameters) (AIEC)|**71.1**|
>
> Experiments show that training the additional linear parameters solely with $\mathcal{L}\_{aux}$ does not improve model performance, and that the actual performance contribution of AIEC comes from $\mathcal{L}\_{gate}$, while introducing $\mathcal{L}\_{gate}$ on top of $\mathcal{L}\_{aux}$ does not add any parameters.
>
> Next, we compare the proposed AIEC with Sparse Coding, $L_0$ Regularization, CBAM, and SENet, which have similar or more parameter counts than AIEC. Sparse coding focuses on obtaining sparse coefficients so that fewer non-zero coefficients are used to represent the original data. $L_0$ Regularization directly penalizes the number of non-zero features to encourage sparsity. CBAM and SENet contain channel or spatial attention. The results on CIFAR-100 are as follows, where "size=" denotes the number of basis vectors:
>
> |Method|Top-1 Acc / % on ViT-Tiny|Top-1 Acc / % on ResNet-18|
> |-|-|-|
> |Sparse Coding (size=100)|66.5|76.1|
> |Sparse Coding (size=500)|65.0|76.2|
> |ReLU + $L_0$ Regularization|67.1|76.5|
> |GELU + $L_0$ Regularization|67.2|76.2|
> |SENet|65.7|75.6|
> |CBAM|N/A (due to its conv layers)|76.2|
> |AIEC|**71.1**|**77.0**|
>
> The results indicate that AIEC's performance gains come from its feature relevance awareness, rather than from solely more parameters.

---

> ### Author Response · Authors · 2025-11-20
> **Author Response to Reviewer g8BT (Part 2/3)**
>
> > **W3:** Writing could be improved: **(a)** The phrase "decision weight" is never defined but it's used extensively including in the abstract. **(b)** Text is duplicated unnecessarily (the caption for Figure 1 is same as text on lines 149-156). **(c)** Figure 2 is referenced on Page 1 long before Figure 1 which isn't referenced until Page 3. **(d)** There is no clear separation of training versus inference in the algorithm description (indeed, there are no algorithm figures at all -- just equations). **(e)** Some equations seem incorrect.
>
> We sincerely appreciate that you raised these points to help us improve our paper and apologize for any confusion caused (a & e).
>
> **(a)** Please refer to our response to **Q1** below.
>
> **(b)** We thank the reviewer for pointing this out. We have rewritten the caption for Figure 1 in the revised manuscript.
>
> **(c)** Our original intention was to include the observation figure in Section 2 ("Observations") for the convenience of readers to read this section. We apologize if this arrangement has caused any inconvenience when reading Section 1 ("Introduction"). We have swapped the positions of Figure 1 and 2 in the revised manuscript.
>
> **(d)** The detailed algorithms are now provided in **Appendix A.5** "Training and Inference Procedures" in the revised manuscript.
>
> **(e)** We believe all equations are correct. Please refer to our response to **Q2** and **Q3** below.
>
> &nbsp;
>
> > **Q1:** What is a "decision weight"? The paper is written seemingly assuming readers are familiar with this term, but I haven't come across it that I can remember.
>
> We are sincerely sorry for the confusion. The decision weights in deep neural networks refer to the weights in the output layer (it outputs predicted scores or others), which are the key parameters relied upon in the final decision-making process of the network. For instance, in classification models, they manifest as the weights of the classification head, while in object detection models, they correspond to the detection head like the box head.
>
> We mentioned the concept of decision weights in some sections of the original manuscript and provided an explicit definition in Line 215:
>
> > > Line 50: Extrinsically, the decision weights learned by a linear classifier acting on the post-activation feature vector can also reflect each channel's relevance to the final decision.
> > >
> > > Caption of Figure 2 (original version) / Figure 1 (revised version): The decision weight is learned through a linear classifier applied to the post-activation feature vector.
> > >
> > > Line 130-134: Observation 2: Linear classifier acting on activated features learns distinct decision weights for each channel, extrinsically indicating channel relevance. ... For classification tasks, we introduce a linear classifier (without the bias term) ...
> > >
> > > **Line 215:** A DMU contains $K$ learnable decision weights $\\{{\mathbf w}^k\\}_{k=1}^K$, respectively for the $K$ categories, where ${\mathbf w}^k\in{\mathbb R}^C$.
>
> Decision weights refer to the weights of a **linear** classifier. The weights of a linear classifier form a matrix $\mathbf{W}$ with the shape [num_features, num_classes]. By performing matrix multiplication with the input features ($\mathbf{y} = \mathbf{W}\mathbf{x}$), it outputs prediction scores corresponding to each class. In Line 215, we split them according to the number of classes ${\\{\mathbf{w}^k\\}}_{k=1}^K$. In Figure 2 (original version) / Figure 1 (revised version), we display the decision weights corresponding to the "truck" class, as mentioned in the caption.
>
> &nbsp;
>
> > **Q2:** What if ${\mathbf a}_c^k<0$  in Equation 4? Is there some reason to believe that Equation 3 should result in non-negative elements of $\mathbf a$?
>
> Thank you for your comment. In the original submission, as we noted in Line 168, the range of values for the post-activation feature map $\mathbf A$ is $[0, +∞]$. Therefore, $\mathbf A$'s pooled vector $\mathbf a$ in Eq. 3 must also be non-negative ( $≥0$ ). As a result, the case ${\mathbf{a}}_c^k < 0$ in Eq. 4 cannot occur.
>
> Going back further, we noted in Line 161 of the original submission that:
>
> > > Line 161: ... given a **non-negative** activation threshold $\tau$, the input signal $x$ is allowed to pass to the next layer as it exceeds the threshold $\tau$, otherwise it gets inhibited to zero. ...
>
> Therefore, with the definition of TAU in Eq. 1 and a non-negative activation threshold $\tau$, the TAU's output $\mathbf A$ in Eq. 2 is non-negative, and so is its pooled vector $\mathbf a$ in Eq. 3.
>
> It is also noteworthy that the activated elements of $\mathbf a$ derived from Eq. 3 and used as the input to Eq. 8 should all be non-negative ( $≥0$ ) to guarantee the interpretability of the proposed method. We put the explanation for this at the end of our response.

---

> ### Author Response · Authors · 2025-11-20
> **Author Response to Reviewer g8BT (Part 3/3)**
>
> > **Q3:** Does Equation 13 not suppress the channels with high relevance? If the point of Equation 11 is to find channels of high relevance, this seems to me to be what would happen with the loss function in Equation 13.
>
> There might be a misunderstanding, and we are glad to clarify this aspect of work.
>
> Eq. 7 and Eq. 10 use the indicator "1" to signify potential ***irrelevant*** channels (also as stated in Line 212 and 232), not relevant channels, **so the point of Eq. 11 is to find channels of *low relevance*** and Eq. 13 only suppresses these channels of low relevance.
>
> The reviewer may refer to our **original submission** for verification. Although we have uploaded a revised version of the paper, these specific parts remain unchanged.
>
> &nbsp;
>
> > **Q4:** Can you outline the steps required to apply your proposal in training and then in inference? I didn't see an algorithm figure in the main text or the appendices.
>
> We truly appreciate this helpful suggestion, and sure, we have provided detailed algorithm descriptions in **Appendix A.5** "Training and Inference Procedures" in the revised manuscript. You can kindly refer to it. We believe it would help you better understand our proposed method.
>
> &nbsp;
>
> ### ***More explanations to Q2***
>
> It is also noteworthy that the activated elements of $\mathbf a$ derived from Eq. 3 and used as the input to Eq. 8 should all be at least non-negative ( $≥0$ ) to guarantee the interpretability of the proposed method. To maintain generality, we denote $\mathbf a$ as $\mathbf{x}$ and provide the following explanation:
>
> The number of channels (feature vector's length) is $C$. In the decision-making process of the model, the predicted score $y$ is typically obtained through the dot product of the input features $\mathbf x=\\{x_1,x_2,\ldots,x_C\\}$ and linear weights $\mathbf w=\\{w_1,w_2,\ldots,w_C\\}$ as $y=\mathbf x \cdot {\mathbf w}^\text{T}$ (Eq. 8). This process can be expressed in the form of weighted summation as $y=w_1x_1+w_2x_2+\ldots+w_Cx_C$. Only when the input feature $x_c$ is positive ($x_c>0$) does its contribution $w_cx_c$ to the score $y$ exhibit a positive correlation with its weight $w_c$. To get a higher score $y$, important features tend to be assigned higher weights to amplify their impact (Eq. 9). Consequently, the learned linear weights $\\{w_1,w_2,\ldots,w_C\\}$ can quantify the relevance of input features $\\{x_1,x_2,\ldots,x_C\\}$ to the network's decision in a one-to-one correspondence (i.e., $w_c$ to $x_c$) (Eq. 10). From a purely symbolic perspective, a positive weight $w_c>0$ implies that the feature $x_c$ contributes positively to the predicted score $y$ (Eq. 10), while a negative weight $w_c<0$ indicates that the feature $x_c$ has a negative contribution to the predicted score $y$ (Eq. 10), in which case suppressing $x_c$ to zero would increase $y$ (Eq. 13, it is another explanation of the gatekeeping).
>
> (Note: $x_c=0$ represents the feature already deemed irrelevant and got inhibited. Its contribution $w_cx_c$ remains zero, and the gradient $\nabla_{w_c}y$ also remains zero. It does not require $w_c$ to further determine relevance and does not cause any updates to $w_c$.)
>
> However, if some $x_c<0$, we cannot determine the relevance of features solely by the sign of the learned $w_c$. If all $x_c$ are non-positive, it is theoretically equivalent to non-negative, but the conclusion is always counterintuitive and violates the concept of "activation".
>
> In summary, the input features $\mathbf x=\\{x_1,x_2,\ldots,x_C\\}$ should all be non-negative ( $≥0$ ) to guarantee the interpretability.
>
> &nbsp;
>
> ***The revised manuscript (PDF) has been uploaded, in which all changes are highlighted in red for your convenience.***
>
> &nbsp;
>
> Thank you once again for your efforts in providing comments that have substantially improved our manuscript. We hope our explanations can address any doubts or concerns you may have, and apologize for any unintended verbosity. If you have any other unclear parts, we welcome your corrections and look forward to more discussions with you!

---

### Official Review · Reviewer_MwHQ · 2025-10-28

**Soundness:** 3
**Presentation:** 3
**Contribution:** 3
**Rating:** 4
**Confidence:** 3

**Summary:**

This study analyzed the channel relevance in Artificial Neural Networks (ANN) and proposed an Activation with Intrisinc-Extrinsic Consensus (AIEC) mechanism that identifies irrelevant channels and performs gatekeeping for noise cleaning. Empirical results demonstrated that the AIEC mechanism outperforms previous activation mechanisms on classification and anomaly detection tasks.

**Strengths:**

1. This study revisited the activation mechanisms and demonstrated that gatekeeping from a global feature perspective may outperform original/traditional activation mechanisms.
2. The proposed AIEC methodology exhibits robust improvements over the baseline and SOTA methods in commonly used image classification benchmarks, and anomaly detection tasks.

**Weaknesses:**

1. While activation based on global feature learning may demonstrate better performance, the mechanism's scalability could be its weakness. It may require higher communication overhead that could affect training efficiency in distributed training, and requries specifc caring for every new architecture. While the additional GPU Memory cost of the proposed AIEC method is neglectable in the scope of this paper, the induced cost may become significant if applied to larger architectures or data dimensions; A theoretical analysis on the computational complexity of AIEC may help to justify.

**Questions:**

1. What is the params size of the ViT-tiny model utilized in this paper? (as their are multiple claimed VIT-tiny models) A more detailed experimental setup section in the appendix is appreciated.
2. As AIEC achieved improvements on Vision Transformers, is it possible to extend the mechanism to language transformer models, in particular GPT-2 with an accessible computational costs (in future studies)?
3. Channels are not a uniform feature in machine learning or deep learning models and tasks; can the authors discuss if the AIEC mechanism can be extended to a different level, for example attention heads, or individual neurons?

---

> ### Author Response · Authors · 2025-11-20
> **Author Response to Reviewer MwHQ (Part 1/2)**
>
> We are grateful to the reviewer for the time and effort spent on reviewing our manuscript and for providing helpful comments.
>
> &nbsp;
>
> > **W1:** While activation based on global feature learning may demonstrate better performance, the mechanism's scalability could be its weakness. It may require higher communication overhead that could affect training efficiency in distributed training, and requires specific caring for every new architecture. While the additional GPU Memory cost of the proposed AIEC method is neglectable in the scope of this paper, the induced cost may become significant if applied to larger architectures or data dimensions; A theoretical analysis on the computational complexity of AIEC may help to justify.
>
> We appreciate the reviewer's attention to detail in noting these points. These suggestions present important opportunities for us to refine and strengthen our work.
>
> &nbsp;
>
> ##  ***Computational complexity***
>
> Theoretically, the time complexity and space complexity of AIEC's Threshold Activation Unit (TAU), Activation-Counting Unit (ACU), Decision-Making Unit (DMU), and Consensus Gatekeeping Unit (CGU) are as follows:
>
> |Training|Threshold Activation Unit (TAU)|Activation-Counting Unit (ACU)|Decision-Making Unit (DMU)|Consensus Gatekeeping Unit (CGU)|
> |-|-|-|-|-|
> |Time Complexity|$O(Input)$|$O(C)$|$O(C\cdot K)$|$O(C)$|
> |Space Complexity|$O(Input)$|$O(C\cdot K)$|$O(C\cdot K)$|$O(C)$|
>
> |Inference|Threshold Activation Unit (TAU)|Activation-Counting Unit (ACU)|Decision-Making Unit (DMU)|Consensus Gatekeeping Unit (CGU)|
> |-|-|-|-|-|
> |Time Complexity|$O(Input)$|not involved|not involved|not involved|
> |Space Complexity|$O(Input)$|not involved|not involved|not involved|
>
> where $Input$ denotes the shape of the input feature map, $C$ denotes the number of feature channels, $K$ denotes the number of categories/classes.
>
> Note that DMU's computation comes from matrix multiplication (Eq. 8) and loss computation, in which the matrix multiplication (Eq. 8) can be highly accelerated through parallel processing. Therefore in practice, although the computational complexity of DMU is $O(C\cdot K)$ while ACU is $O(C)$, the actual time cost of DMU is much smaller than that of ACU. The average time taken by each part of AIEC in the ViT-Tiny model (the sum of 12 layers) to process a batch of data (1024 images with 224×224-pixels) on an NVIDIA A6000 GPU is as follows:
>
> |AIEC|Threshold Activation Unit (TAU)|Activation-Counting Unit (ACU)|Decision-Making Unit (DMU)|Consensus Gatekeeping Unit (CGU)|
> |-|-|-|-|-|
> |Training time per batch|3.282ms|100.303ms|4.706ms|12.048ms|
>
> ACU actually has the highest time overhead, while DMU has the second lowest, even close to the fastest TAU. In addition, we can further reduce the time cost of ACU (Eq. 4-6) and DMU (Eq. 8-9) by lowering their update frequency (e.g., updating every 5 iterations).
>
> &nbsp;
>
> ##  ***Scalability to larger models***
>
> ### *(1) Communication overhead in distributed training*
>
> **It is not necessary to introduce cross-device communication for AIEC in distributed training.** The statistical information of ACU can be maintained independently on a per-device basis, the parameter updates of DMU can be limited to the local device, and CGU contains no information that requires synchronization. The explanation is as follows:
>
> During the inference phase, ACU, DMU, and CGU are all discarded, leaving only the parameter-free TAU part. This indicates that the information from ACU, DMU, and CGU has already been incorporated into other parts of the model through gradient backpropagation, so only the other parts of the model need to synchronize information across devices. Therefore, AIEC can completely avoid cross-device communication during training.
>
> ### *(2) Adaptation costs on new architectures*
>
> **The code of AIEC has been encapsulated and can be easily integrated into new architectures.** We have designed our AIEC to be adaptable to various ANN architectures including CNNs, ViTs, and GNNs, which is not difficult to implement.
>
> ### *(3) GPU memory costs*
>
> **The proportion of GPU overhead from AIEC tends to become smaller when applied to larger-scale models.** For example, the comparison of GPU memory when training ViT-Tiny, ViT-Base, and ViT-Huge is as follows. The models were fed 224×224-pixel images with a batch size of 64 on an NVIDIA A6000 GPU.
>
> |Training GPU Memory|ViT-Tiny|ViT-Base|ViT-Huge|
> |-|-|-|-|
> |ReLU|2.374 GiB|8.008 GiB|43.642 GiB|
> |AIEC|2.420 GiB|8.136 GiB|43.978 GiB|
> |$\Delta$|1.9%|1.6%|0.8%|
>
> Possible reasons are that:
>
> - As the embedding dimension $C$ of the model increases, the computational cost of modules like self-attention grows quadratically ($O(C^2)$), while the computational cost of our AIEC increases linearly ($O(C)$).
>
> - As the model scale increases, other complex modules may be introduced, further reducing the proportion of AIEC's GPU memory costs.

---

> ### Author Response · Authors · 2025-11-20
> **Author Response to Reviewer MwHQ (Part 2/2)**
>
> > **Q1:** What is the params size of the ViT-tiny model utilized in this paper? (as there are multiple claimed VIT-tiny models) A more detailed experimental setup section in the appendix is appreciated.
>
> Thank you for your valuable comments. The params size of the ViT-Tiny model is 5 M. ViT-Tiny is a standard model architecture defined in the commonly used `timm` library. We have provided a more detailed experimental setup section in **Appendix A.6**, including model settings and experimental settings. Please kindly refer to it.
> |Argument|Value|
> |-|-|
> |input image size|32 for CIFAR / 224 for ImageNet|
> |patch size| 4 for CIFAR / 16 for ImageNet   |
> |embed dim|192|
> |# layers|12|
> |# heads|3|
> |# params|5 M|
>
> &nbsp;
>
> > **Q2:** As AIEC achieved improvements on Vision Transformers, **(a)** is it possible to extend the mechanism to language transformer models, **(b)** in particular GPT-2 with an accessible computational costs (in future studies)?
>
> We sincerely appreciate your insightful comment.
>
> **(a)** Yes, the mechanism can be extended to language transformer models. For text classification, we evaluate the proposed AIEC on the BERT (Devlin et al., 2019) model with the 20News dataset. The results show that AIEC remains effective.
>
> |Method|Top-1 Acc / %|
> |-|-|
> |BERT w/ ReLU|86.7|
> |BERT w/ GELU|86.9|
> |BERT w/ AIEC|**88.0**|
>
> **(b)** It is also possible to extend AIEC to GPT-2 or other LLMs **because their next-token prediction is essentially the classification task** where the number of classes is the size of the vocabulary.
>
> **On the computational costs**, referring to our response to **W1**:
>
> - For larger-scale models, the proportion of the proposed AIEC tends to become smaller. This is because, as the embedding dimension of the model increases, the computational cost of modules like self-attention grows quadratically ($O(C^2)$), while the computational complexity of our AIEC increases linearly ($O(C)$).
>
> - Additionally, as the model scale increases, other complex modules may be introduced, further reducing the proportion of AIEC's GPU memory costs.
> - Moreover, the extra computation introduced by AIEC occurs only in the training phase. During inference, only the basic Threshold Activation Unit (TAU) is involved, so the inference overhead remains nearly unchanged.
>
> &nbsp;
>
> > **Q3:** Channels are not a uniform feature in machine learning or deep learning models and tasks; can the authors discuss if the AIEC mechanism can be extended to a different level, for example attention heads, or individual neurons?
>
> Your thoughtful comment is much appreciated. We discuss this point as follows:
>
> The global feature of a model is typically represented in the form of a feature vector, and channels are referred to as the vector's elements. For example, Vision Transformer uses a class token, while CNNs globally pool feature maps into a feature vector. Therefore, applying AIEC to the channels in a global feature vector can help directly improve model performance.
>
> The essence of AIEC is to distinguish between relevant and irrelevant features within a set of features and suppress the irrelevant ones. In theory, AIEC can be applied to any form of features, such as feature vectors, attention heads, or individual neurons. However, the issue lies in the granularity:
>
> - **Applying AIEC to channels of a feature vector:** The global feature vector of a model typically contains around 1000 channels, which is a suitable scale for feature selection with only modest computational costs and additional parameters.
> - **Applying AIEC to attention heads:** This granularity is somewhat too coarse. A model usually has only 1 to 16 attention heads, and fully suppressing even a few of them may lead to performance degradation. However, for larger models with more attention heads, head-wise gating may also be effective.
> - **Applying AIEC to individual neurons:** This granularity is somewhat too fine. The number of neurons is generally extremely large, and maintaining statistics and learning weights for each neuron would incur significant computational overhead and too many parameters. Gating at the channel level may already effectively capture most of the redundant/irrelevant information. The performance improvement achieved by operating at a finer granularity may not justify the additional computations and parameters.
>
> Therefore, considering the trade-off between computational costs and performance, it is most appropriate to apply AIEC at the channel level.
>
> &nbsp;
>
> ***The revised manuscript (PDF) has been uploaded, in which all changes are highlighted in red for your convenience.***
>
> &nbsp;
>
> Thank you once again for your time in reviewing our paper. We hope that our revisions and responses will be found satisfactory, and we look forward to hearing from you!

---

### Official Review · Reviewer_hHGn · 2025-10-29

**Soundness:** 2
**Presentation:** 3
**Contribution:** 2
**Rating:** 4
**Confidence:** 3

**Summary:**

This paper proposes AIEC (Activation with Intrinsic-Extrinsic Consensus), a novel activation mechanism that identifies and suppresses irrelevant feature channels during neural network training through a consensus-based assessment combining intrinsic activity statistics and extrinsic decision weights. The core observation is that channel relevance can be assessed from both the frequency of channel activation (intrinsic) and learned classifier weights (extrinsic), and these two assessments exhibit strong consensus. The method is evaluated across multiple architectures (ViTs, CNNs, GNNs) and datasets (CIFAR-10/100, ImageNet-100/1K, graph datasets).This paper proposes AIEC (Activation with Intrinsic-Extrinsic Consensus), a novel activation mechanism that identifies and suppresses irrelevant feature channels during neural network training through a consensus-based assessment combining intrinsic activity statistics and extrinsic decision weights. The core observation is that channel relevance can be assessed from both the frequency of channel activation (intrinsic) and learned classifier weights (extrinsic), and these two assessments exhibit strong consensus. The method is evaluated across multiple architectures (ViTs, CNNs, GNNs) and datasets (CIFAR-10/100, ImageNet-100/1K, graph datasets).

**Strengths:**

- The method's effectiveness is demonstrated across a diverse set of architectures, including multiple ViT variants, standard CNNs (AlexNet, VGG, ResNet, etc.), and even Graph Neural Networks (GCN, GraphSAGE).
- The inclusion of non-vision graph datasets (T-Finance, Elliptic, Weibo) to test generalization beyond computer vision is a commendable strength.
- The proposed intrinsic-extrinsic duality provides a clear and intuitive structure for conceptualizing the problem of channel relevance.
- The proposed mechanism (ACU, DMU, CGU) operates only during training, meaning it adds no computational or latency overhead at inference time, as shown in Table 4.

**Weaknesses:**

- This is the most significant flaw. AIEC introduces two substantial auxiliary loss terms ($\mathcal{L}_{aux}$, $\mathcal{L}_{gate}$) and a complex stateful counting mechanism. It is then compared against simple, stateless activation functions like ReLU and GELU. A fair evaluation would require comparing AIEC to other stateful channel selection/gating mechanisms or, at minimum, adding equivalent regularization to the baselines.
- The core idea of using auxiliary losses (extrinsic) or activation statistics (intrinsic) to gauge importance is well-established. The novelty hinges entirely on their consensus, yet the empirical justification for this added complexity is weak. The ablation study (Table 3) shows the intrinsic-only approach ($AIEC_I$) achieves 69.7% accuracy, while the full consensus model ($AIEC_{I\cap E}$) achieves 71.1%. The paper fails to provide a strong justification for why this modest 1.4% gain is worth the significant increase in mechanism complexity.
- (Fig. 2, 3) The paper provides no quantitative correlation analysis between the intrinsic and extrinsic assessments to substantiate this claim.
- The method is applied inconsistently across architectures without a clear justification. The authors apply AIEC to every block in Transformer models but only to the last block in CNNs.
- While the paper claims negligible overhead, Table 4 shows a non-trivial 9.5% increase in training latency for ViT-Tiny. But, its too small models and datasets to support your word.
- The Threshold Activation Unit (TAU) ultimately employs a fixed threshold of zero, rendering it functionally identical to ReLU. The experimental results presented in the Appendix merely reaffirm the known effectiveness of ReLU, rather than demonstrating any distinct contribution of TAU itself. Consequently, TAU does not offer any genuine novelty within this work and can be regarded as a rephrased formulation of ReLU.

**Questions:**

- Could you please provide a comparison against other baselines with (a) comparing AIEC to other dynamic channel selection or gating mechanisms, or (b) re-running baseline (ReLU, GELU)?
- (Table 3) Ablation shows only 1.4% gain from consensus vs. intrinsic alone. can you provide a stronger theoretical or empirical justification for the added complexity of the DMU and the consensus mechanism?
- The ACU and DMU both scale with the number of classes, $K$. Could the authors provide a FLOPs analysis and a memory/latency scalability study for much larger $K$?
- Why apply to all ViT blocks but only last CNN block? Could you provide systematic layer-wise application study?
- What makes channels irrelevant? Are they truly noisy or just task-irrelevant?
- Since AIEC identifies irrelevant channels, can you prune them post-training? What accuracy-efficiency tradeoffs result?

---

> ### Author Response · Authors · 2025-11-20
> **Author Response to Reviewer hHGn (Part 1/4)**
>
> Thank you very much for your careful review and thoughtful comments, which provide us with important guidance for refining our work.
>
> &nbsp;
>
> > **W1:** This is the most significant flaw. AIEC introduces two substantial auxiliary loss terms ($\mathcal{L}\_{aux}$, $\mathcal{L}\_{gate}$) and a complex stateful counting mechanism. It is then compared against simple, stateless activation functions like ReLU and GELU. A fair evaluation would require comparing AIEC to other stateful channel selection/gating mechanisms or, at minimum, adding equivalent regularization to the baselines.
>
> We appreciate the reviewer's thoughtful comment. The proposed AIEC addresses the shortcoming (the lack of feature relevance awareness) in existing activation functions such as ReLU and GELU, representing a new form of activation mechanism. Thus, we compare AIEC with these mainstream static activation functions to demonstrate the performance gain it brings.
>
> We agree that adding more comparisons would help further improve the robustness of our evaluation. Accordingly, we compare the proposed AIEC with:
>
> - **Equivalent regularization ($L_0$ Regularization)**: It directly penalizes the number of non-zero features to encourage sparsity. However, when the number of features is large, such a constraint makes it difficult for the model to determine which specific elements should be suppressed to zero. In contrast, our proposed AIEC provides clear criteria indicating which elements should be suppressed, offering more sufficient supervision signals and stronger interpretability.
> - **Dynamic channel selection or gating mechanisms (CBAM and SENet)**: They introduce additional learnable FC or Conv layers to generate attention weights at the channel level (or the spatial level). It simply weights the channels **without** evidence of sparse representation or interpretability. In contrast, the proposed AIEC leverages both intrinsic and extrinsic characteristics of threshold activation as direct supervision signals to identify irrelevant channels and apply noise cleaning targeted on them, thus providing strong sparsity and interpretability.
>
> The comparison results on CIFAR-100 are as follows, demonstrating that AIEC still holds a clear performance advantage.
>
> |Method|Top-1 Acc / % on ViT-Tiny|Top-1 Acc / % on ResNet-18|
> |-|-|-|
> |ReLU|65.7|75.7|
> |GELU|65.4|75.6|
> |ReLU + $L_0$ Regularization|67.1|76.5|
> |GELU + $L_0$ Regularization|67.2|76.2|
> |SENet (channel selection/gating)|65.7|75.6|
> |CBAM (channel selection/gating)|N/A (due to its conv layers)|76.2|
> |AIEC|**71.1**|**77.0**|
>
> &nbsp;
>
> > **W2:** **(a)** The core idea of using auxiliary losses (extrinsic) or activation statistics (intrinsic) to gauge importance is well-established. The novelty hinges entirely on their consensus, **(b)** yet the empirical justification for this added complexity is weak. The ablation study (Table 3) shows the intrinsic-only approach ($AIEC_I$) achieves 69.7% accuracy, while the full consensus model ($AIEC_{I\cap E}$) achieves 71.1%. The paper fails to provide a strong justification for why this modest 1.4% gain is worth the significant increase in mechanism complexity.
>
> We are sincerely sorry for any confusion caused.
>
> **(a)** **The extrinsic and intrinsic parts have our unique contributions to this work.** The extrinsic part distinguishes between irrelevant and relevant channels based on the sign of a linear classifier's weights. The intrinsic part maintains three sets of counters for historical activation states and introduces a "relative firing rate" metric to reflect channel relevance. These aspects have not been addressed in other works.
>
> **(b)** **Actually, integration of the extrinsic part does not lead to a significant increase in mechanism complexity.** Here is the clarification on this issue:
>
> The computation of the extrinsic part comes from matrix multiplication (Eq. 8) and loss computation, in which **the matrix multiplication (Eq. 8) can be highly accelerated through parallel processing**. Therefore in practice, although the computational complexity of the extrinsic part is $O(C\cdot K)$ while the intrinsic part is $O(C)$, **the actual time cost of the extrinsic part is much smaller than that of the intrinsic part.** The average time taken by each part of AIEC in the ViT-Tiny model (the sum of 12 layers) to process a batch of data (1024 images with 224×224-pixels) on an NVIDIA A6000 GPU is as follows:
>
> |AIEC|Threshold Activation Unit (TAU)|Activation-Counting Unit (ACU, the intrinsic part)|Decision-Making Unit (DMU, the extrinsic part)|Consensus Gatekeeping Unit (CGU)|
> |-|-|-|-|-|
> |Training time per batch|3.282ms|100.303ms|4.706ms|12.048ms|
>
> The intrinsic part actually has the highest time overhead, while the extrinsic part has the second lowest, even close to the fastest TAU part. In conclusion, the extrinsic part does not cause a significant increase in mechanism complexity, which makes it well worth the performance gain it delivers.

---

> ### Author Response · Authors · 2025-11-20
> **Author Response to Reviewer hHGn (Part 2/4)**
>
> > **W3:** (Fig. 2, 3) The paper provides no quantitative correlation analysis between the intrinsic and extrinsic assessments to substantiate this claim.
>
> We greatly value your constructive feedback.
>
> Based on Figure 2(b1)(b2) of the original submission (or Figure 1(b1)(b2) of the revised manuscript), denote the vector recording activity levels (b1, blue) as $\mathbf v$ and the vector recording decision weights (b2, green) as ${\mathbf w}$. They both contain 768 channels. We provide the following quantitative metrics:
>
> - **Pearson correlation coefficient** between $\mathbf v$ and $\mathbf w$, which ranges from -1 (perfect negative correlation) to 1 (perfect positive correlation), emphasizing a linear relationship.
> - **Spearman correlation coefficient** between $\mathbf v$ and $\mathbf w$, which ranges from -1 (perfect negative correlation) to 1 (perfect positive correlation), emphasizing a monotonic relationship.
> - **Average intrinsic score** of $\mathbf v$ and **Average extrinsic score** of $\mathbf w$ under different channel subsets $S$. The lower the score, the less relevant the channel.
>
> The Pearson and Spearman correlation coefficients between $\mathbf v$ (b1, blue) and $\mathbf w$ (b2, green) are as follows, which shows a strong **positive correlation**, providing solid quantitative evidence for our claim of "consensus".
>
> |Pair|Pearson Corr.|Spearman Corr.|
> |-|-|-|
> |$\mathbf v$ & $\mathbf w$|0.5549|0.5202|
>
> Moreover, average intrinsic and extrinsic scores under different channel subsets are as follows, which further demonstrate **the high consistency between intrinsic and extrinsic views** in identifying irrelevant channels and that identifying irrelevant channels **under intrinsic-extrinsic consensus is better than that under a single view**.
>
> |Channel Subset ($S$)|Avg. Intrinsic Score ( $\frac{1}{\|S\|}\sum{\mathbf v}_{i\in S}$ )|Avg. Extrinsic Score ( $\frac{1}{\|S\|}\sum{\mathbf w}_{i\in S}$ )|
> |-|-|-|
> |All channels|-0.2813|-0.0237|
> |Irrelevant by intrinsic only ( ${\mathcal G}^{k, Intr}$, Eq. 7 )|-0.3751|-0.0392|
> |Irrelevant by extrinsic only ( ${\mathcal G}^{k, Extr}$, Eq. 10 )|-0.3962|-0.0750|
> |Irrelevant by consensus ( ${\mathcal G}^{k, Intr}\cap{\mathcal G}^{k, Extr}$, Eq. 11 )|-0.4104|-0.0762|
> |Relevant by consensus ( $\neg{\mathcal G}^{k, Intr}\cap\neg{\mathcal G}^{k, Extr}$, Eq. 11 )|+0.2593|+0.0741|
>
> Additionally, in Figure 3, the proportions of active channels (${\mathbf v}_i>0$) before and after using AIEC are as follows, implying that irrelevant channels are effectively identified and suppressed.
>
> ||w/o AIEC|w/ AIEC|
> |-|-|-|
> |Proportion of Active Channels|14.84 %|7.68 %|
>
> &nbsp;
> > **W4:** The method is applied inconsistently across architectures without a clear justification. The authors apply AIEC to every block in Transformer models but only to the last block in CNNs.
>
> We are sincerely sorry for the confusion. As we stated in Line 260-262 of the original submission,
>
> > > Line 260-262: ... for Transformer models, the proposed AIEC is applied to each block, as Transformers excel in capturing global context throughout, while for CNN models, AIEC is applied to the last block since high-level semantics only exist in deep representations (Raghu et al., 2021).
>
> Previous research [1] found that ViTs exhibit a more uniform representational structure, with high representational similarity across all layers. In contrast, CNNs show lower similarity between shallow-layer and deep-layer representations, where shallow layers capture local features and only deep-layer representations contain high-level semantics.
>
> *[1] Raghu et al., Do vision transformers see like convolutional neural networks? NeurIPS 2021.*
>
> Since our proposed AIEC operates on global feature vectors, the class token at any layer of a ViT can represent global features, whereas in CNNs, only deep-layer features contain high-level information. Features in the shallow layers of CNNs are too concrete. Therefore, imposing the gatekeeping to a channel (an element) cannot directly control the change in the corresponding spatial-level feature map.
>
> Moreover, we have explored the impact of applying AIEC to different layers of ResNet-18 on CIFAR-100 as follows, where L4 is the last layer:
>
> |ResNet-18 Layer(s)|Top-1 Acc / %| ResNet-18 Layer(s) |Top-1 Acc / %|
> |-|-|-|-|
> |L1|75.8|L1-3|76.0|
> |L2|75.6|L3-4|76.9|
> |L3|76.2|L2-4|76.5|
> |L4|77.0|L1-4|76.8|
>
> The results indicate that for CNNs like ResNet-18, applying AIEC to shallow layers has little effect, and only constraining the last layer can achieve the optimal effect.
>
> Similar experiments are conducted on ViT-Tiny as follows, where L12 is the last layer:
>
> |ViT-Tiny Layer(s)|Top-1 Acc / %| ViT-Tiny Layer(s) |Top-1 Acc / %|
> |-|-|-|-|
> |L1|66.3|L1,4,8,12|67.5|
> |L4|66.6|L1-6|68.2|
> |L8|67.0|L7-12|68.9|
> |L12|66.7|L1-12|71.1|
>
> The results indicate that for ViTs like ViT-Tiny, applying AIEC to shallow or deep layers has similar effects, and the more layers AIEC is applied to, the better the performance.

---

> ### Author Response · Authors · 2025-11-20
> **Author Response to Reviewer hHGn (Part 3/4)**
>
> > **W5:** While the paper claims negligible overhead, Table 4 shows a non-trivial 9.5% increase in training latency for ViT-Tiny. But, it's too small models and datasets to support your word.
>
> We sincerely apologize for any confusion caused. Actually, we did not exaggerate to claim "negligible overhead" **but it is "negligible GPU overhead"** as we stated in Line 323 of our original submission.
>
> > > Line 323: Table 4 shows that AIEC's **training GPU overhead is negligible** and its inference speed is on par with other methods.
>
> Regarding the issue itself, first, the training latency can be further reduced by **decreasing the update frequency** of ACU (Eq. 4-6) and DMU (Eq. 8-9) (e.g., updating every 10 iterations).
>
> Next, **to further examine how training latency scales with model size**, the comparison of latency when training ViT-Tiny, ViT-Base, and ViT-Huge is as follows. The models were fed ImageNet-1K (224×224-pixel images) with a batch size of 64 on an NVIDIA A6000 GPU.
>
> |Training Latency|ViT-Tiny|ViT-Base|ViT-Huge|
> |-|-|-|-|
> |ReLU|73.3 ms|294.0 ms|1123.0 ms|
> |AIEC|80.5 ms|315.2 ms|1201.6 ms|
> |$\Delta$|9.8%|7.2%|7.0%|
>
> The results indicate that the proportion of training latency from AIEC tends to decrease when applied to larger-scale models. Potential reasons for this trend include: (i) As the embedding dimension $C$ of the model increases, the computational cost of modules like self-attention grows quadratically ($O(C^2)$), while the computational cost of our AIEC increases linearly ($O(C)$). (ii) As the model scale increases, other complex modules may be introduced, further reducing the proportion of AIEC's training latency.
>
> &nbsp;
>
> > **W6:** **(a)** The Threshold Activation Unit (TAU) ultimately employs a fixed threshold of zero, rendering it functionally identical to ReLU. **(b)** The experimental results presented in the Appendix merely reaffirm the known effectiveness of ReLU, rather than demonstrating any distinct contribution of TAU itself. **(c)** Consequently, TAU does not offer any genuine novelty within this work and can be regarded as a rephrased formulation of ReLU.
>
> We are happy to provide clarification on this point.
>
> **(a)** The Threshold Activation Unit (TAU) we defined serves as a generalized form of "threshold firing", where the threshold $\tau$ can take any non-negative value, and ReLU is merely a special case of TAU when $\tau=0$. The optimal performance at $\tau=0$ may be influenced by the "zero-mean" strategy in weight initialization and feature normalization (Line 656-659). This can be altered, as we stated in Line 660:
>
> > > Line 660: ... Additionally, it may be feasible to modify the strategy for weight initialization and feature normalization to achieve optimal effects when considering a positive $\tau$.
>
> **(b)** The experiment in Appendix A.1 discusses the impact of different activation thresholds, which was not mentioned in the original work of ReLU, as it just fixed the threshold at zero.
>
> **(c)** Actually, we did not present TAU as a genuine novelty or contribution of our paper as we did not include this in the contribution statement (Line 87-98) or the abstract, but only briefly described it in Section 3.1. **The true contributions of the paper lie in the observation and design of intrinsic and extrinsic channel assessments and the channel gatekeeping based on their consensus.** As a basic module of AIEC, TAU provides a **prior** partition boundary (activation threshold) to divide feature channels into irrelevant and relevant groups. Building upon this, intrinsic and extrinsic channel relevance assessments are derived respectively through activation-counting and decision-making processes in a data-driven manner. Finally, a **posterior** partition boundary between irrelevant and relevant channels is established through intrinsic-extrinsic consensus, enabling channel-wise gatekeeping that effectively suppresses irrelevant channels.
>
> &nbsp;
>
> > **Q1:** Could you please provide a comparison against other baselines with (a) comparing AIEC to other dynamic channel selection or gating mechanisms, or (b) re-running baseline (ReLU, GELU)?
>
> Thank you once again. Please refer to our response to **W1** above.
>
> &nbsp;
>
> > **Q2:** (Table 3) Ablation shows only 1.4% gain from consensus vs. intrinsic alone. Can you provide a stronger theoretical or empirical justification for the added complexity of the DMU and the consensus mechanism?
>
> We apologize again for the confusion. Please refer to our response to **W2** above.

---

> ### Author Response · Authors · 2025-11-20
> **Author Response to Reviewer hHGn (Part 4/4)**
>
> > **Q3:** The ACU and DMU both scale with the number of classes, $K$. Could the authors provide a FLOPs analysis and a memory/latency scalability study for much larger $K$?
>
> We sincerely appreciate your insightful feedback. ACU and DMU primarily involve simple matrix addition and multiplication operations (dimensions include $K$). Theoretically, the time complexity and space complexity of AIEC's Activation-Counting Unit (ACU) and Decision-Making Unit (DMU) are as follows:
>
> |Training|Activation-Counting Unit (ACU)|Decision-Making Unit (DMU)|
> |-|-|-|
> |Space Complexity|$O(C\cdot K)$|$O(C\cdot K)$|
> |Time Complexity|$O(C)$|$O(C\cdot K)$|
>
> |Inference|Activation-Counting Unit (ACU)|Decision-Making Unit (DMU)|
> |-|-|-|
> |Space Complexity|not involved|not involved|
> |Time Complexity|not involved|not involved|
>
> where $C$ denotes the number of feature channels, $K$ denotes the number of classes.
>
> Here we present how the proposed AIEC scales on ViT-Tiny w.r.t. the number of classes $K$ in terms of FLOPs, training GPU memory, and training latency. The ViT-Tiny model was fed 224×224-pixel images with a batch size of 1024 on an NVIDIA A6000 GPU.
>
> |$K$|FLOPs|Training GPU Memory|Training Latency|
> |-|-|-|-|
> |10|1.07844 G|32.152 GiB|589.7 ms|
> |100|1.07846 G|32.174 GiB|591.8 ms|
> |1000|1.07863 G|32.542 GiB|595.1 ms|
> |10000|1.08036 G|34.844 GiB|615.2 ms|
>
> The results indicate that the number of classes $K$ has little impact on the computational cost, demonstrating AIEC's excellent scalability w.r.t. $K$.
>
> &nbsp;
>
> > **Q4:** Why apply to all ViT blocks but only last CNN block? Could you provide systematic layer-wise application study?
>
> Thank you once again. Please refer to our response to **W4** above.
>
> &nbsp;
>
> > **Q5: (a)** What makes channels irrelevant? **(b)** Are they truly noisy or just task-irrelevant?
>
> Thank you for your insightful questions.
>
> **(a)** **Channel relevance is measured based on its contribution to the network's decision.** Taking the image classification task as an example, if a channel's response leads to a higher predicted score for the target class, then that channel is considered relevant. Conversely, if a channel's response lowers the network's predicted score for the target class, then that channel is deemed irrelevant.
>
> **(b)** **They are both task-irrelevant and noisy.** It is precisely because noisy responses from task-irrelevant channels lower the network's prediction score for the target class that we need to suppress these channels to obtain higher prediction scores.
>
> &nbsp;
>
> > **Q6:** Since AIEC identifies irrelevant channels, can you prune them post-training? What accuracy-efficiency tradeoffs result?
>
> This is an interesting comment. Actually, we can't, and more importantly, **don't need to prune them post-training**.
>
> ### *Reasons why we can't prune them post-training*
>
> - The channel relevance assessment of our proposed AIEC is category-specific. That is, for each category, the identified irrelevant channels are different. After training, as the category of test data is unknown, it is not possible to determine which channels are irrelevant for a given test sample.
>
> - As we discussed in Appendix A.2 of the original submission, the target of channel pruning is **channel weights** ${\mathbf W}$ (e.g., convolutional kernels or linear weights), while the target of our AIEC's channel gatekeeping is **channel responses** $\text{output}=f(\text{input};{\mathbf W})$ (e.g., $\text{output}=\text{input}\cdot {\mathbf W}^{\text T}$). It is impossible to prune a response ($\text{output}$) like pruning a weight (${\mathbf W}$).
>
> ### *Reasons why we don't need to prune them post-training*
>
> - During training, the responses from irrelevant channels are suppressed under the guidance of the gatekeeping loss $\mathcal{L}_{gate}$ (Eq. 13). **By the end of training, the model has already learned how to avoid generating responses on irrelevant channels.** Hence, during inference, ACU, DMU, and CGU are all discarded, and only TAU needs to be involved and the model can achieve higher testing accuracy. For details, please refer to the algorithm description provided in **Appendix A.5** of our paper.
>
> &nbsp;
>
> ***The revised manuscript (PDF) has been uploaded, in which all changes are highlighted in red for your convenience.***
>
> &nbsp;
>
> Thank you once again for your guidance. We hope our rebuttal meets the required standards. If you have further questions, please do not hesitate to let us know. We are more than willing to provide further elaboration!

---

> > ### Comment · Reviewer_hHGn · 2025-11-28
> >
> > I would like to thank the authors for their detailed response and for making significant efforts to address my concerns during the rebuttal phase.
> >
> >   In my initial review, I raised a concern regarding the fairness of comparisons. Since AIEC introduces stateful tracking mechanisms and auxiliary loss functions, comparing it solely against simple stateless activation functions (e.g., ReLU, GELU) seemed insufficient, as the performance gains could potentially be attributed to the increased model complexity or parameter count.
> >
> >   However, the authors have effectively addressed this by providing new comparisons with relevant gating and sparsity techniques. I believe these additional results convincingly demonstrate that the performance improvement of AIEC stems from the validity of the proposed mechanism itself, rather than mere parameter inflation.
> >
> >   While I share the sentiment with another reviewer that the theoretical grounding for the optimality of the Consensus mechanism could be further strengthened, I find that the method has been sufficiently validated through the extensive empirical experiments provided.
> >
> >   In light of the robust validation and the authors' constructive response, I am pleased to raise my final rating to 6.

---

> > > ### Author Response · Authors · 2025-11-28
> > > **Many thanks for your recognition and for raising the score to 6.**
> > >
> > > Dear Reviewer hHGn,
> > >
> > > Many thanks for your recognition and for raising the score to 6.
> > >
> > > Your suggestions were highly constructive. We are delighted that our rebuttal and additional experiments, which provide new comparisons with relevant gating and sparsity techniques, have adequately addressed your concerns regarding the fairness of comparisons.
> > >
> > > Also, as you have acknowledged, the extensive empirical experiments we have provided in the original submission, revision, and rebuttal have been sufficient to validate our method.
> > >
> > > Once again, we deeply appreciate your acknowledgment of our work and are truly grateful for the time and effort you have dedicated to reviewing our paper!
> > >
> > > &nbsp;
> > >
> > > Best regards,
> > >
> > > Authors of Paper #1653

---

### Official Review · Reviewer_jQCx · 2025-11-11

**Soundness:** 3
**Presentation:** 3
**Contribution:** 2
**Rating:** 4
**Confidence:** 4

**Summary:**

This paper proposes AIEC (Activation with Intrinsic-Extrinsic Consensus), a novel activation mechanism designed to suppress irrelevant neural channels by leveraging a dual assessment of channel relevance. The method combines intrinsic statistics from activation frequency and extrinsic decision weights learned via a linear classifier, using their consensus to gate unimportant channels during training. Experiments across various architectures (CNNs, ViTs, GNNs) and datasets (CIFAR, ImageNet, Weibo) show that AIEC improves accuracy and sparsity with minimal overhead. While the idea is intuitive and well-implemented, it overlaps conceptually with prior work on feature attribution and sparsity, lacks theoretical grounding for its gating mechanism, and omits comparisons to closely related structured pruning baselines.

**Strengths:**

1. __Clear Identification of a Practical Problem__: The paper addresses a meaningful limitation of existing activation functions, leading to unnecessary activations, which may propagate noise. This is a well-motivated problem with practical implications for sparsity, interpretability, and robustness.

2. __Conceptually Intuitive and Modular Design__: The proposed AIEC framework clearly separates intrinsic (activity-based) and extrinsic (decision-based) assessments of channel relevance, combining them via a consensus mechanism. Each module (ACU, DMU, CGU) has a well-defined role, and the design is simple enough to be adapted across architectures.

3. __Proper Clarity and Visualization__: The paper is well-structured and readable, with well-crafted figures (e.g., Figure 1, Figure 2, and Appendix visualizations) that help explain both the motivation and effects of AIEC. These visualizations qualitatively support the gating effect on irrelevant channels.

4. __Computational Efficiency Report__: The authors provide detailed reporting of GPU memory usage and latency (Table 4), showing that AIEC introduces minimal overhead during training and inference, which indicates an important factor for practical deployment.

**Weaknesses:**

__1. Lack of Theoretical Justification__

One of the primary limitations of the paper is the absence of theoretical background and justification behind its core mechanisms, particularly the consensus-based gating strategy in Eq. (11). While the intuition that irrelevant channels should be identified when both intrinsic (activation-based) and extrinsic (decision-weight-based) assessments agree is reasonable, the choice of a hard intersection as the gating criterion lacks formal justification. There is no information-theoretic, statistical, or optimization-based rationale for why the intersection is better than, for instance, union or threshold-weighted approaches. Additionally, the choice of using a fixed threshold τ = 0 in both the ACU and DMU components is empirically motivated, yet its suitability across architectures and datasets is neither theoretically analyzed nor justified through sensitivity experiments. As a result, the reliability and generalizability of the proposed mechanism remain questionable. However, this concern could be mitigated if the authors provide clear theoretical reasoning or empirical evidence supporting these design choices.


__2. Non-Differentiable Gating and Optimization Concerns__

The gatekeeping mechanism employed in AIEC introduces hard, binary decisions for channel suppression (Eqs. 7, 10, and 12), which are non-differentiable. While the authors state that AIEC operates during training, they do not address how gradient flow is handled through the gating logic. This is particularly concerning as the consensus gate (CGU) functions as a multiplicative mask, effectively blocking gradient propagation for suppressed channels. In the absence of surrogate gradients or soft approximations, this may lead to convergence instability or ineffective training, especially in deeper networks or more complex tasks. The lack of discussion around training dynamics, stability, or gradient handling in the presence of hard gates is a significant omission, raising doubts about the robustness of the proposed method under various optimization conditions. Introducing a differentiable approximation or continuous relaxation of the gating function could address this issue by enabling gradient flow during training [1]. It would be beneficial for the authors to provide justification or analysis of this design choice, especially regarding whether such approximations were considered or why hard gating was preferred.

[1] Lee, Kyungsu, et al. "Stochastic adaptive activation function." Advances in Neural Information Processing Systems 35 (2022): 13787-13799.


__3. Missing Comparisons to Key Baselines__

Can the authors elaborate on the rationale for comparing AIEC only against conventional activation functions such as ReLU, GELU, and SiLU, rather than including more structurally relevant baselines such as L0 regularization, channel pruning techniques, or attention-based gating modules like CBAM[2]? Given that the primary goal of AIEC is to suppress task-irrelevant channels, these methods share a similar motivation and mechanism. Without such comparisons, it becomes unclear whether the observed improvements are due to better activation dynamics or simply the addition of a gating mechanism. Additionally, since the current evaluation focuses mainly on classification tasks, further clarification on whether task-specific constraints or computational considerations drove this choice would help clarify the intended generalizability of the method.

[2] Woo, Sanghyun, et al. "Cbam: Convolutional block attention module." Proceedings of the European conference on computer vision (ECCV). 2018.

**Questions:**

1. __On the Use of a Linear Classifier in the DMU__: The Decision-Making Unit (DMU) employs a linear classifier without a bias term to compute decision weights for each channel. Could the authors clarify why a linear model was chosen, and whether more expressive classifiers (e.g., MLPs or non-linear heads) were considered? How sensitive is the method’s performance to this modeling choice?


2. __On the Epoch-Level Resetting of ACU Statistics__: The Activation-Counting Unit (ACU) resets its statistics at the beginning of each epoch, as stated in Section 3.4. What is the rationale behind this design choice? Have the authors evaluated the impact of accumulating statistics across epochs, and whether that would lead to more stable or reliable intrinsic relevance estimates?

---

> ### Author Response · Authors · 2025-11-20
> **Author Response to Reviewer jQCx (Part 1/3)**
>
> We thank the reviewer for the insightful comments and constructive suggestions, which have greatly helped us to improve the manuscript.
>
> &nbsp;
>
> > **W1:** Lack of Theoretical Justification
> >
> > **(a)** On the intersection criterion (Eq. 11)
> >
> > **(b)** On the threshold $\tau$
>
> We appreciate the reviewer for raising these points.
>
> **On the intersection criterion (Eq. 11):** Due to the presence of noise and errors, the channel relevance assessment cannot be 100% accurate. Only suppressing channels that are deemed irrelevant by both intrinsic and extrinsic assessments aims to minimize misjudgments. This is a cautious and conservative strategy (rather let go than wrongfully suppress). As shown in Table 3, the intersection-based approach ($AIEC_{I\cap E}$) outperforms both single-assessment ($AIEC_I$, $AIEC_E$) and the union-based variant ($AIEC_{I\cup E}$), which can support its practical effectiveness. Nevertheless, we acknowledge the importance of theoretical grounding and will consider incorporating formal theoretical studies in our future research.
>
> **On the threshold $\tau$:** Appendix A.1 of our original submission discussed the reason to choose $\tau=0$, which is applicable across architectures and datasets:
>
> > > Line 655-660: Possible reasons could be the influence of weight initialization and feature normalization operations. Typically, weights are initialized using a distribution with a mean of zero, and normalization techniques such as layer normalization and batch normalization are used to make the feature distribution centered around zero (by subtracting the feature mean) to eliminate shifts in data covariates. Under these circumstances, $\tau=0$ becomes the optimal activation threshold. Additionally, it may be feasible to modify the strategy for weight initialization and feature normalization to achieve optimal effects when considering a positive $\tau$.
>
> Figure 4 of our original submission presents the sensitivity experiments of $\tau$ when training on CIFAR-100 with ViT-Tiny.
>
> Here, we supplement sensitivity experiments of $\tau$ across architectures and datasets:
>
> |Top-1 Acc (%)|$\tau=0$|$\tau=0.4$|$\tau=1$|$\tau=10$|
> |-|-|-|-|-|
> |ViT-Tiny & CIFAR-10|91.8|78.3|74.8|73.2|
> |ViT-Tiny & CIFAR-100|71.2|67.1|55.8|53.5|
> |ResNet-18 & CIFAR-10|95.1|88.6|88.0|46.9|
> |ResNet-18 & CIFAR-100|77.0|76.5|75.8|64.8|
>
> The results consistently indicate that $\tau=0$ is optimal.
>
> &nbsp;
>
> > **W2:** Non-Differentiable Gating and Optimization Concerns.
> >
> > **(a)** The gatekeeping mechanism employed in AIEC introduces hard, binary decisions for channel suppression (**Eqs. 7, 10, and 12**), **which are non-differentiable**. ...
> >
> > **(b)** This is particularly concerning as the consensus gate (CGU) functions as a **multiplicative mask**, effectively **blocking gradient propagation for *suppressed* channels**. ...
>
> We sincerely regret any confusion caused. There might be some misunderstandings as:
>
> **(a)** The gatekeeping mechanism in AIEC is **differentiable**.
>
> - **Eq. 7, 10:** The binary 0/1 mask ${\mathcal G}^k$ itself is indeed not updated by gradient descent. It is designed to be determined by the intrinsic metric $\mathbf v$ and the extrinsic metric $\mathbf w$, which update dynamically during training.
>
> - **Eq. 12:** The operation ${\check{\mathbf a}}^k={\mathbf a}^k\odot {\mathcal G}^k$ is fully differentiable to ${\mathbf a}^k$, as it is the element-wise multiplication (Hadamard Product $\odot$) of the activated feature vector ${\mathbf a}^k$ and the mask ${\mathcal G}^k$, where ${\mathcal G}^k$ is just a **constant** mask (in the view of gradient calculation) to select irrelevant channels.
>
> **(b)** The multiplicative mask **${\mathcal G}^k$** (in Eq. 12) does **not** "block gradient propagation for *suppressed* channels", but blocks gradient propagation for *unsuppressed* channels while allowing gradients to propagate through suppressed channels.
>
> The multiplicative mask ${\mathcal G}^k$ in Eq. 12 is used to **select** irrelevant channels, rather than suppress them. The actual channel suppression takes place in Eq. 13, conducting on the **selected** irrelevant channels via loss constraint ${\mathcal L}_{gate}$. As a result, the gradients from Eq. 13 that suppresses irrelevant information can propagate in the channels that require suppression, while in the channels that should be preserved (do not require suppression), the gradient propagation is blocked (means the gradients are zero, but does not mean it is non-differentiable) by the mask ${\mathcal G}^k$ in Eq. 12.
>
> Additionally, based on the reasoning above, the differential approximation or continuous relaxation in reference [1] you mentioned is not necessary in our method, and the training in our study is **stable**.
>
> (Please note that the reviewer referred to ${\mathcal G}^k$ as a "mask", but we did not refer to ${\mathcal G}^k$ as so in our paper. Instead, we called it "channel relevance division", which is used to **select** but not to mask.)

---

> ### Author Response · Authors · 2025-11-20
> **Author Response to Reviewer jQCx (Part 2/3)**
>
> > **W3:** Missing Comparisons to Key Baselines
> >
> > **(a)** Can the authors elaborate on the rationale for comparing AIEC only against conventional activation functions such as ReLU, GELU, and SiLU, rather than including more structurally relevant baselines such as L0 regularization, channel pruning techniques, or attention-based gating modules like CBAM[2]? ... **(b)** Additionally, since the current evaluation focuses mainly on classification tasks, further clarification on whether task-specific constraints or computational considerations drove this choice would help clarify the intended generalizability of the method.
>
> We appreciate your valuable feedback.
>
> **(a)** The proposal of AIEC was intended to benchmark against existing activation mechanisms, for the following reasons:
>
> - **Motivation:** Existing activation mechanisms like ReLU and GELU are static, with a simple and fixed decision boundary to separate relevant from irrelevant features during training, lacking interpretable evidence for feature relevance awareness. The proposed AIEC is a novel activation mechanism with feature relevance awareness, capable of dynamically adjusting and optimizing the decision boundary between irrelevant and relevant features based on intrinsic-extrinsic consensus, leading to more precise activation. By addressing the shortcomings of existing activation mechanisms, the proposed AIEC should represent a new form of activation.
> - **Objective/Effect:** The proposed AIEC aims to achieve sparse feature representations by inhibiting irrelevant features. This aligns with the objective/effect of activation functions such as ReLU.
> - **Application:** The proposed AIEC is implemented in code by replacing the code of existing activation functions.
>
> Discussion on other baselines:
>
> - **$L_0$ regularization:** $L_0$ regularization constrains the number of non-zero elements in features by adding a loss term. However, when the number of features is large, such a constraint makes it difficult for the model to determine which specific elements should be suppressed to zero. In contrast, our proposed AIEC provides clear criteria indicating which elements should be suppressed, offering more sufficient supervision signals and stronger interpretability.
> - **Channel pruning:** As we discussed in Appendix A.2 of the original submission, the target of channel pruning is **channel weights** ${\mathbf W}$, while the target of our proposed channel gatekeeping is **channel responses** $f(\text{input};{\mathbf W})$. The objects they refer to as "channel" are different. Furthermore, to the best of our knowledge, channel pruning techniques by reducing the number of model parameters to achieve higher computational efficiency are typically accompanied by a decrease in model accuracy.
> - **Attention-based gating modules:** Modules like SENet and CBAM introduce additional learnable FC or Conv layers to generate attention weights at the channel level (or the spatial level). It simply weights the channels **without** evidence of sparse representation or interpretability. In contrast, the proposed AIEC leverages both intrinsic and extrinsic characteristics of threshold activation as direct supervision signals to identify irrelevant channels and apply noise cleaning targeted on them, thus providing strong sparsity and interpretability.
>
> Here we also compare AIEC with these baselines as follows:
>
> |Method|Top-1 Acc / % on ViT-Tiny|Top-1 Acc / % on ResNet-18|
> |-|-|-|
> |ReLU + $L_0$ Regularization|67.1|76.5|
> |GELU + $L_0$ Regularization|67.2|76.2|
> |Network Slimming (channel pruning)|N/A (it is for CNNs)|74.5|
> |SENet (attention-based gating)|65.7|75.6|
> |CBAM (attention-based gating)|N/A (due to its conv layers)|76.2|
> |AIEC|**71.1**|**77.0**|
>
> The results demonstrate that the proposed AIEC achieves superior performance compared to these methods. We have incorporated these comparisons in **Appendix A.7** of the revised manuscript.
>
> **(b)**  We chose the classification task as the primary evaluation benchmark, primarily based on the following considerations:
>
> - It provides a pure and unbiased verification scenario. Classification models typically do not involve complex structures, allowing improvements in model performance to be most directly attributed to enhanced feature representation quality, while also facilitating interpretability analysis.
> - Broad applicability:
>   - Pre-trained backbones from classification tasks can be directly applied to many downstream tasks like object detection, semantic segmentation, etc.
>   - For currently popular Large Language Models (LLMs), their next-token prediction is essentially the classification task where the number of classes is the size of the vocabulary.
> - Its high computational efficiency makes it suitable for rapid validation and obtaining robust conclusions under limited computational resources.
>
> Therefore, the intended generalizability of the proposed AIEC across various other tasks (especially in LLMs) can be anticipated.

---

> ### Author Response · Authors · 2025-11-20
> **Author Response to Reviewer jQCx (Part 3/3)**
>
> > **Q1:** On the Use of a Linear Classifier in the DMU: **(a)** The Decision-Making Unit (DMU) employs a linear classifier without a bias term to compute decision weights for each channel. Could the authors clarify why a linear model was chosen, and whether more expressive classifiers (e.g., MLPs or non-linear heads) were considered? **(b)** How sensitive is the method's performance to this modeling choice?
>
> Thank you for your insightful comment.
>
> **(a)** This touches upon a crucial point: **Linear models are highly interpretable**, which can be used with positive (or non-negative) inputs for interpretability analysis.
>
> The number of channels (feature vector's length) is $C$. In Decision-Making Unit (DMU), the predicted score $y$ is typically obtained through the dot product of the input features $\mathbf x=\\{x_1,x_2,\ldots,x_C\\}$ (i.e., the $\mathbf a$ in Eq. 3 & 8) and linear weights $\mathbf w=\\{w_1,w_2,\ldots,w_C\\}$ as $y=\mathbf x \cdot {\mathbf w}^\text{T}$ (Eq. 8). This process can be expressed in the form of weighted summation as $y=w_1x_1+w_2x_2+\ldots+w_Cx_C$. With positive input feature $x_c$ ($x_c>0$), its contribution $w_cx_c$ to the score $y$ exhibits a positive correlation with its weight $w_c$. To get a higher score $y$, important features tend to be assigned higher weights to amplify their impact (Eq. 9). Consequently, the learned linear weights $\\{w_1,w_2,\ldots,w_C\\}$ can quantify the relevance of input features $\\{x_1,x_2,\ldots,x_C\\}$ to the network's decision in a one-to-one correspondence (i.e., $w_c$ to $x_c$) (Eq. 10). From a purely symbolic perspective, a positive weight $w_c>0$ implies that the feature $x_c$ contributes positively to the predicted score $y$ (Eq. 10), while a negative weight $w_c<0$ indicates that the feature $x_c$ has a negative contribution to the predicted score $y$ (Eq. 10), in which case suppressing $x_c$ to zero would increase $y$ (Eq. 13, it is another explanation of the gatekeeping).
>
> In contrast, **more complex classifiers such as MLPs or non-linear heads do not preserve such interpretability and are more like "black boxes"**. Therefore, what we need is not a highly expressive classifier, but rather an interpretable one, whose weights can reflect the relevance of input features to the network's decision in a one-to-one correspondence (i.e., $w_c$ to $x_c$).
>
> **(b)** Moreover, since the weights must correspond one-to-one with the elements of the input feature vector (i.e., $w_c$ to $x_c$), the length of weights must be $C$ to match the feature vector's length, and also since the channel relevance assessment is category-specific that requires distinct weights to be learned for each category ($K$ categories in total), **the weights of shape $C×K$ is the only option, which is a linear classifier. Alternatives such as MLPs or non-linear heads are not applicable in this context.** Even if they were applicable, their non-linear factors would make the conclusion less reliable.
>
> &nbsp;
>
> > **Q2:** On the Epoch-Level Resetting of ACU Statistics: **(a)** The Activation-Counting Unit (ACU) resets its statistics at the beginning of each epoch, as stated in Section 3.4. What is the rationale behind this design choice? **(b)** Have the authors evaluated the impact of accumulating statistics across epochs, and whether that would lead to more stable or reliable intrinsic relevance estimates?
>
> We thank the reviewer for highlighting this point.
>
> **(a)** The purpose of resetting the counters in ACU at the beginning of each epoch is to guarantee the timeliness and accuracy of statistics (Line 263-264), which can prevent the accumulation of potential historical statistical biases.
>
> **(b)** In practice, we found that accumulating statistics across epochs can also achieve nearly the same performance. The possible reason is that during the early stages of training, as the model converges rapidly, a small amount of noise is quickly "diluted" by statistics gathered in subsequent training phases. Theoretically, **statistics reset at the beginning of each epoch are more reliable**, while **statistics accumulated across epochs are more stable**. In this paper, since we tend to obtain more reliable samples, we chose the former strategy.
>
> &nbsp;
>
> ***The revised manuscript (PDF) has been uploaded, in which all changes are highlighted in red for your convenience.***
>
> &nbsp;
>
> Thank you once again for your time and effort in reviewing our paper. We hope our rebuttal will address your concerns. We are glad to have further discussions with you!

---

### Author Response · Authors · 2025-12-02
**Author Response to Area Chair (AC)**

&nbsp;
## **Dear AC,**
Thank you for your valuable time!

With full respect for you and all the reviewers, we would like to provide some explanations to avoid any potential misunderstandings before you make a final decision.

&nbsp;
## **Brief Summary**
Overall, the reviewers did not question the paper's novelty or contributions.

They only pointed out some experiments that need to be supplemented, as well as details of the proposed method and some unclear aspects that require further clarification.

The concerns they raised are all easy to address.

&nbsp;
## ***Reviewer jQCx (No Response)***
> **W1:** Lack of Theoretical Justification
>
>(on the intersection criterion (Eq. 11) and the threshold $\tau$)

Apart from our explanations in the rebuttal, we have provided extensive empirical experiments in the original submission, revision, and rebuttal. These have been sufficient to validate our method, which has been acknowledged by **Reviewer hHGn (Score 4→6)** in the post-rebuttal phase.

&nbsp;
> **W2:** Non-Differentiable Gating and Optimization Concerns

The reviewer misunderstood our method to be non-differentiable and unstable for optimization. However, the fact is that:

**(a)** The operation ${\check{\mathbf a}}^k={\mathbf a}^k\odot {\mathcal G}^k$ (**Eq. 12**) is fully **differentiable** to ${\mathbf a}^k$, as it is the element-wise multiplication (Hadamard Product $\odot$) of the activated feature vector ${\mathbf a}^k$ and the binary 0/1 mask ${\mathcal G}^k$. The ${\mathcal G}^k$ (**Eq. 7, 10**) is just a **constant** mask (in the view of gradient calculation) to select irrelevant channels. It is designed to be determined by the intrinsic metric $\mathbf v$ and the extrinsic metric $\mathbf w$, updating dynamically during training.

**(b)** The ${\mathcal G}^k$ (in Eq. 12) does **not**, as the reviewer said, "block gradient propagation for *suppressed* channels", but blocks gradient propagation for *unsuppressed* channels while allowing gradients to propagate through suppressed channels. The use of ${\mathcal G}^k$ is to **select** irrelevant channels, rather than suppress them. The actual channel suppression takes place in Eq. 13, conducting on the **selected** irrelevant channels via loss constraint ${\mathcal L}_{gate}$, ensuring stable training.

&nbsp;
> **W3:** Missing Comparisons to Key Baselines

We have provided detailed discussions and comparisons to other structurally relevant baselines. **Reviewer hHGn (Score 4→6)**, who originally raised this concern also, is now satisfied with our response and considers the issue resolved.

&nbsp;
> **Q1:** On the Use of a Linear Classifier in the DMU
>
> **Q2:** On the Epoch-Level Resetting of ACU Statistics

We believe we have explained these issues clearly. Please see our rebuttal.

&nbsp;
## ***Reviewer hHGn (4→6)***
Our rebuttal has received the reviewer's recognition, which addressed a wide range of questions that overlapped with those from other reviewers.

&nbsp;
## ***Reviewer MwHQ (No Response)***
The reviewer did not raise any sharp questions, whose main concern is on the method's computational complexity and scalability to larger models.

We have provided thorough analyses and discussions demonstrating that our method does not introduce much computational and memory overhead when applied to larger models. We believe this should address the reviewer's main concern.

&nbsp;
## ***Reviewer g8BT (No Response)***
The reviewer has raised some highly insightful points and provided valuable suggestions for improving our paper in the "Weaknesses" part, for which we are very grateful.

However, the reviewer appears **not to have fully understood our paper**, particularly by overlooking some methodological details and misinterpreting some of our key equations.

For instance,
> **Q3:** Does Equation 13 not suppress the channels with *high relevance*? If the point of Equation 11 is to find channels of *high relevance*, this seems to me to be what would happen with the loss function in Equation 13.

The reviewer may have misinterpreted and reversed the meaning in Eq. 7 and Eq. 10.

Eq. 7 and Eq. 10 use the indicator "1" to signify potential ***irrelevant*** channels (also as stated in Line 212 and 232), **not** relevant channels, **so the point of Eq. 11 is to find channels of *low relevance*** and Eq. 13 only suppresses these channels of low relevance.

***We must emphasize that this is exactly how we presented it in our original submission, and we did not make any revisions to these parts.***

We have also supplemented detailed training and inference algorithms as requested in **Appendix A.5**, as it will help facilitate a better understanding of our method.

&nbsp;

&nbsp;

We sincerely appreciate and fully respect all the reviewers' constructive feedback and provide these essential explanations to you to avoid any potential misunderstandings. We apologize for any unintended verbosity. Thank you once again!

&nbsp;

Sincerely,

Authors of Paper #1653

---

### Meta-Review · Area_Chair_nBLQ · 2026-01-06

**Summary:**

Overall, several concerns were raised by the reviewers, ranging from lack of proper theoretical justification or rather positioning with respecting to existing ideas (jQCx, g8BT and hHGn), optimization concerns (jQCx), missing comparisons (jQCx and g8BT), lack of quantitative correlation analysis (hHGn), unmatched claims (hHGn), concerns related to scalability (MwHQ). The evaluation before rebuttal is unanimously moderately negative.

**Reviewer Concerns:**

After the very long rebuttal, the optimization concerns are clarified, together with a better positioning with respect to scalability and correction to unmatched claims. Also some comparisons are provided for jQCx. However, the positioning with respect to existing ideas, the grounding and theoretical justification of the chosen approach still remains unclear. Together with this, the answer provided to g8BT lacks theoretical depth and requires major rework of the whole manuscript to be addressed.

**Reviewer Scores:**

Overall, some reviewers would have increased their score, but still remaining for the majority leaning to reject the work. The work is acknowledged to be interesting, but still requires major work to be ready for publication.

---

### Decision · Program_Chairs · 2026-01-26

Reject